# ProMP: Proximal Meta-Policy Search

**Jonas Rothfuss**[*]
UC Berkeley, KIT
jonas.rothfuss@gmail.com

**Dennis Lee**[*]**, Ignasi Clavera**[*]
UC Berkeley
{dennisl88,iclavera}@berkeley.edu

**Tamim Asfour**
Karlsruhe Inst. of Technology (KIT)
asfour@kit.edu

**Pieter Abbeel**
UC Berkeley, Covariant.ai
pabbeel@cs.berkeley.edu

## ABSTRACT

Credit assignment in Meta-reinforcement learning (Meta-RL) is still poorly understood. Existing methods either neglect credit assignment to pre-adaptation behavior or implement it naively. This leads to poor sample-efficiency during meta-training as well as ineffective task identification strategies. This paper provides a theoretical analysis of credit assignment in gradient-based Meta-RL. Building on the gained insights we develop a novel meta-learning algorithm that overcomes both the issue of poor credit assignment and previous difficulties in estimating meta-policy gradients. By controlling the statistical distance of both pre-adaptation and adapted policies during meta-policy search, the proposed algorithm endows efficient and stable meta-learning. Our approach leads to superior pre-adaptation policy behavior and consistently outperforms previous Meta-RL algorithms in sample-efficiency, wall-clock time, and asymptotic performance.

## 1 INTRODUCTION

A remarkable trait of human intelligence is the ability to adapt to new situations in the face of limited experience. In contrast, our most successful artificial agents struggle in such scenarios. While achieving impressive results, they suffer from high sample complexity in learning even a single task, fail to generalize to new situations, and require large amounts of additional data to successfully adapt to new environments. Meta-learning addresses these shortcomings by learning how to learn. Its objective is to learn an algorithm that allows the artificial agent to succeed in an unseen task when only limited experience is available, aiming to achieve the same fast adaptation that humans possess (Schmidhuber, 1987; Thrun & Pratt, 1998).

Despite recent progress, deep reinforcement learning (RL) still relies heavily on hand-crafted features and reward functions as well as engineered problem specific inductive bias. Meta-RL aims to forego such reliance by acquiring inductive bias in a data-driven manner. Recent work proves this approach to be promising, demonstrating that Meta-RL allows agents to obtain a diverse set of skills, attain better exploration strategies, and learn faster through meta-learned dynamics models or synthetic returns (Duan et al., 2016; Xu et al., 2018; Gupta et al., 2018b; Saemundsson et al., 2018).

Meta-RL is a multi-stage process in which the agent, after a few sampled environment interactions, adapts its behavior to the given task. Despite its wide utilization, little work has been done to promote theoretical understanding of this process, leaving Meta-RL grounded on unstable foundations. Although the behavior prior to the adaptation step is instrumental for task identification, the interplay between pre-adaptation sampling and posterior performance of the policy remains poorly understood. In fact, prior work in gradient-based Meta-RL has either entirely neglected credit assignment to the pre-update distribution (Finn et al., 2017) or implemented such credit assignment in a naive way (Al-Shedivat et al., 2018; Stadie et al., 2018).

To our knowledge, we provide the first formal in-depth analysis of credit assignment w.r.t. pre-adaptation sampling distribution in Meta-RL. Based on our findings, we develop a novel Meta-RL algorithm. First, we analyze two distinct methods for assigning credit to pre-adaptation behavior.

---

[*]authors contributed equally to this work

We show that the recent formulation introduced by Al-Shedivat et al. (2018) and Stadie et al. (2018) leads to poor credit assignment, while the MAML formulation (Finn et al., 2017) potentially yields superior meta-policy updates. Second, based on insights from our formal analysis, we highlight both the importance and difficulty of proper meta-policy gradient estimates. In light of this, we propose the low variance curvature (LVC) surrogate objective which yields gradient estimates with a favorable bias-variance trade-off. Finally, building upon the LVC estimator we develop Proximal Meta-Policy Search (ProMP), an efficient and stable meta-learning algorithm for RL. In our experiments, we show that ProMP consistently outperforms previous Meta-RL algorithms in sample-efficiency, wall-clock time, and asymptotic performance.

## 2 RELATED WORK

Meta-Learning concerns the question of "learning to learn", aiming to acquire inductive bias in a data driven manner, so that the learning process in face of unseen data or new problem settings is accelerated (Schmidhuber, 1987; Schmidhuber et al., 1997; Thrun & Pratt, 1998).

This can be achieved in various ways. One category of methods attempts to learn the "learning program" of an universal Turing machine in form of a recurrent / memory-augmented model that ingests datasets and either outputs the parameters of the trained model (Hochreiter et al., 2001; Andrychowicz et al., 2016; Chen et al., 2017; Ravi & Larochelle, 2017) or directly outputs predictions for given test inputs (Duan et al., 2016; Santoro et al., 2016; Mishra et al., 2018). Though very flexible and capable of learning very efficient adaptations, such methods lack performance guarantees and are difficult to train on long sequences that arise in Meta-RL.

Another set of methods embeds the structure of a classical learning algorithm in the meta-learning procedure, and optimizes the parameters of the embedded learner during the meta-training (Hüsken & Goerick, 2000; Finn et al., 2017; Nichol et al., 2018; Miconi et al., 2018). A particular instance of the latter that has proven to be particularly successful in the context of RL is gradient-based meta-learning (Finn et al., 2017; Al-Shedivat et al., 2018; Stadie et al., 2018). Its objective is to learn an initialization such that after one or few steps of policy gradients the agent attains full performance on a new task. A desirable property of this approach is that even if fast adaptation fails, the agent just falls back on vanilla policy-gradients. However, as we show, previous gradient-based Meta-RL methods either neglect or perform poor credit assignment w.r.t. the pre-update sampling distribution.

A diverse set of methods building on Meta-RL, has recently been introduced. This includes: learning exploration strategies (Gupta et al., 2018b), synthetic rewards (Sung et al., 2017; Xu et al., 2018), unsupervised policy acquisition (Gupta et al., 2018a), model-based RL (Clavera et al., 2018; Saemundsson et al., 2018), learning in competitive environments (Al-Shedivat et al., 2018) and meta-learning modular policies (Frans et al., 2018; Alet et al., 2018). Many of the mentioned approaches build on previous gradient-based meta-learning methods that insufficiently account for the pre-update distribution. ProMP overcomes these deficiencies, providing the necessary framework for novel applications of Meta-RL in unsolved problems.

## 3 BACKGROUND

**Reinforcement Learning.** A discrete-time finite Markov decision process (MDP), $\mathcal{T}$, is defined by the tuple $(\mathcal{S}, \mathcal{A}, p, p_0, r, H)$. Here, $\mathcal{S}$ is the set of states, $\mathcal{A}$ the action space, $p(s_{t+1}|s_t, a_t)$ the transition distribution, $p_0$ represents the initial state distribution, $r : \mathcal{S} \times \mathcal{A} \to \mathbb{R}$ is a reward function, and $H$ the time horizon. We omit the discount factor $\gamma$ in the following elaborations for notational brevity. However, it is straightforward to include it by substituting the reward by $r(s_t, a_t) := \gamma^t r(s_t, a_t)$. We define the return $R(\tau)$ as the sum of rewards along a trajectory $\tau := (s_0, a_0, ..., s_{H-1}, a_{H-1}, s_H)$. The goal of reinforcement learning is to find a policy $\pi(a|s)$ that maximizes the expected return $\mathbb{E}_{\tau \sim P_{\mathcal{T}}(\tau|\pi)}[R(\tau)]$.

**Meta-Reinforcement Learning** goes one step further, aiming to learn a learning algorithm which is able to quickly learn the optimal policy for a task $\mathcal{T}$ drawn from a distribution of tasks $\rho(\mathcal{T})$. Each task $\mathcal{T}$ corresponds to a different MDP. Typically, it is assumed that the distribution of tasks share the action and state space, but may differ in their reward function or their dynamics.

**Gradient-based meta-learning** aims to solve this problem by learning the parameters $\theta$ of a policy $\pi_\theta$ such that performing a single or few steps of vanilla policy gradient (VPG) with the given task leads to the optimal policy for that task. This meta-learning formulation, also known under the name

of MAML, was first introduced by Finn et al. (2017). We refer to it as formulation I which can be expressed as maximizing the objective

$$J^I(\theta) = \mathbb{E}_{\mathcal{T} \sim \rho(\mathcal{T})} \big[ \mathbb{E}_{\boldsymbol{\tau}' \sim P_{\mathcal{T}}(\boldsymbol{\tau}'|\theta')} \left[ R(\boldsymbol{\tau}') \right] \big] \quad \text{with} \quad \theta' := U(\theta, \mathcal{T}) = \theta + \alpha \nabla_\theta \mathbb{E}_{\boldsymbol{\tau} \sim P_{\mathcal{T}}(\boldsymbol{\tau}|\theta)} \left[ R(\boldsymbol{\tau}) \right]$$

In that $U$ denotes the update function which depends on the task $\mathcal{T}$, and performs one VPG step towards maximizing the performance of the policy in $\mathcal{T}$. For national brevity and conciseness we assume a single policy gradient adaptation step. Nonetheless, all presented concepts can easily be extended to multiple adaptation steps.

Later work proposes a slightly different notion of gradient-based Meta-RL, also known as E-MAML, that attempts to circumvent issues with the meta-gradient estimation in MAML (Al-Shedivat et al., 2018; Stadie et al., 2018):

$$J^{II}(\theta) = \mathbb{E}_{\mathcal{T} \sim \rho(\mathcal{T})} \big[ \mathbb{E}_{\substack{\boldsymbol{\tau}^{1:N} \sim P_{\mathcal{T}}(\boldsymbol{\tau}^{1:N}|\theta) \\ \boldsymbol{\tau}' \sim P_{\mathcal{T}}(\boldsymbol{\tau}'|\theta')}} \big[ R(\boldsymbol{\tau}') \big] \big] \text{ with } \theta' := U(\theta, \boldsymbol{\tau}^{1:N}) = \theta + \alpha \nabla_\theta \sum_{n=1}^{N} \Big[ R(\boldsymbol{\tau}^{(n)}) \Big]$$

Formulation II views $U$ as a deterministic function that depends on $N$ sampled trajectories from a specific task. In contrast to formulation I, the expectation over pre-update trajectories $\boldsymbol{\tau}$ is applied outside of the update function. Throughout this paper we refer to $\pi_\theta$ as pre-update policy, and $\pi_{\theta'}$ as post-update policy.

## 4 SAMPLING DISTRIBUTION CREDIT ASSIGNMENT

This section analyzes the two gradient-based Meta-RL formulations introduced in Section 3. Figure 1 illustrates the stochastic computation graphs (Schulman et al., 2015b) of both formulations. The red arrows depict how credit assignment w.r.t the pre-update sampling distribution $P_{\mathcal{T}}(\boldsymbol{\tau}|\theta)$ is propagated. Formulation I (left) propagates the credit assignment through the update step, thereby exploiting the full problem structure. In contrast, formulation II (right) neglects the inherent structure, directly assigning credit from post-update return $R'$ to the pre-update policy $\pi_\theta$ which leads to noisier, less effective credit assignment.

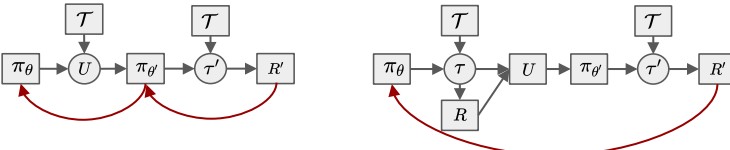

Figure 1: Stochastic computation graphs of meta-learning formulation I (left) and formulation II (right). The red arrows illustrate the credit assignment from the post-update returns $R'$ to the pre-update policy $\pi_\theta$ through $\nabla_\theta J_{\text{pre}}$. (Deterministic nodes: Square; Stochastic nodes: Circle)

Both formulations optimize for the same objective, and are equivalent at the $0^{th}$ order. However, because of the difference in their formulation and stochastic computation graph, their gradients and the resulting optimization step differs. In the following, we shed light on how and where formulation II loses signal by analyzing the gradients of both formulations, which can be written as (see Appendix A for more details and derivations)

$$\nabla_\theta J(\theta) = \mathbb{E}_{\mathcal{T} \sim \rho(\mathcal{T})} \left[ \mathbb{E}_{\substack{\boldsymbol{\tau} \sim P_{\mathcal{T}}(\boldsymbol{\tau}|\theta) \\ \boldsymbol{\tau}' \sim P_{\mathcal{T}}(\boldsymbol{\tau}'|\theta')}} \left[ \nabla_\theta J_{\text{post}}(\boldsymbol{\tau}, \boldsymbol{\tau}') + \nabla_\theta J_{\text{pre}}(\boldsymbol{\tau}, \boldsymbol{\tau}') \right] \right] \tag{1}$$

The first term $\nabla_\theta J_{\text{post}}(\boldsymbol{\tau}, \boldsymbol{\tau}')$ is equal in both formulations, but the second term, $\nabla_\theta J_{\text{pre}}(\boldsymbol{\tau}, \boldsymbol{\tau}')$, differs between them. In particular, they correspond to

$$\nabla_\theta J_{\text{post}}(\boldsymbol{\tau}, \boldsymbol{\tau}') = \underbrace{\nabla_{\theta'} \log \pi_\theta(\boldsymbol{\tau}') R(\boldsymbol{\tau}')}_{\nabla_{\theta'} J^{\text{outer}}} \underbrace{\left( I + \alpha R(\boldsymbol{\tau}) \nabla_\theta^2 \log \pi_{\theta'}(\boldsymbol{\tau}) \right)}_{\text{transformation from } \theta' \text{ to } \theta} \tag{2}$$

$$\nabla_\theta J_{\text{pre}}^{II}(\boldsymbol{\tau}, \boldsymbol{\tau}') = \alpha \nabla_\theta \log \pi_\theta(\boldsymbol{\tau}) R(\boldsymbol{\tau}') \tag{3}$$

$$\nabla_\theta J_{\text{pre}}^{I}(\boldsymbol{\tau}, \boldsymbol{\tau}') = \alpha \nabla_\theta \log \pi_\theta(\boldsymbol{\tau}) \bigg( \underbrace{\left( \nabla_\theta \log \pi_\theta(\boldsymbol{\tau}) R(\boldsymbol{\tau}) \right)^\top}_{\nabla_\theta J^{\text{inner}}} \underbrace{\left( \nabla_{\theta'} \log \pi_{\theta'}(\boldsymbol{\tau}') R(\boldsymbol{\tau}') \right)}_{\nabla_{\theta'} J^{\text{outer}}} \bigg) \tag{4}$$

$\nabla_\theta J_{\text{post}}(\boldsymbol{\tau}, \boldsymbol{\tau}')$ simply corresponds to a policy gradient step on the post-update policy $\pi_{\theta'}$ w.r.t $\theta'$, followed by a linear transformation from post- to pre-update parameters. It corresponds to increasing the likelihood of the trajectories $\boldsymbol{\tau}'$ that led to higher returns. However, this term does not optimize for the pre-update sampling distribution, i.e., which trajectories $\boldsymbol{\tau}$ led to better adaptation steps.

The credit assignment w.r.t. the pre-updated sampling distribution is carried out by the second term. In formulation II, $\nabla_\theta J_{\text{pre}}^{II}$ can be viewed as standard reinforcement learning on $\pi_\theta$ with $R(\boldsymbol{\tau}')$ as reward signal, treating the update function $U$ as part of the unknown dynamics of the system. This shifts the pre-update sampling distribution to better adaptation steps.

Formulation I takes the causal dependence of $P_\mathcal{T}(\boldsymbol{\tau}'|\theta')$ on $P_\mathcal{T}(\boldsymbol{\tau}|\theta)$ into account. It does so by maximizing the inner product of pre-update and post-update policy gradients (see Eq. 4). This steers the pre-update policy towards 1) larger post-updates returns 2) larger adaptation steps $\alpha\nabla_\theta J^{\text{inner}}$, 3) better alignment of pre- and post-update policy gradients (Li et al., 2017; Nichol et al., 2018). When combined, these effects directly optimize for adaptation. As a result, we expect the first meta-policy gradient formulation, $J^I$, to yield superior learning properties.

## 5  LOW VARIANCE CURVATURE ESTIMATOR

In the previous section we show that the formulation introduced by Finn et al. (2017) results in superior meta-gradient updates, which should in principle lead to improved convergence properties. However, obtaining correct and low variance estimates of the respective meta-gradients proves challenging. As discussed by Foerster et al. (2018), and shown in Appendix B.3, the score function surrogate objective approach is ill suited for calculating higher order derivatives via automatic differentiation toolboxes. This important fact was overlooked in the original RL-MAML implementation (Finn et al., 2017) leading to incorrect meta-gradient estimates[1]. As a result, $\nabla_\theta J_{\text{pre}}$ does not appear in the gradients of the meta-objective (i.e. $\nabla_\theta J = \nabla_\theta J_{\text{post}}$). Hence, MAML does not perform any credit assignment to pre-adaptation behavior.

But, even when properly implemented, we show that the meta-gradients exhibit high variance. Specifically, the estimation of the hessian of the RL-objective, which is inherent in the meta-gradients, requires special consideration. In this section, we motivate and introduce the low variance curvature estimator (LVC): an improved estimator for the hessian of the RL-objective which promotes better meta-policy gradient updates. As we show in Appendix A.1, we can write the gradient of the meta-learning objective as

$$\nabla_\theta J^I(\theta) = \mathbb{E}_{\mathcal{T}\sim\rho(\mathcal{T})}\Big[\mathbb{E}_{\boldsymbol{\tau}'\sim P_\mathcal{T}(\boldsymbol{\tau}'|\theta')}\big[\nabla_{\theta'}\log P_\mathcal{T}(\boldsymbol{\tau}'|\theta')R(\boldsymbol{\tau}')\nabla_\theta U(\theta, \mathcal{T})\big]\Big] \tag{5}$$

Since the update function $U$ resembles a policy gradient step, its gradient $\nabla_\theta U(\theta, \mathcal{T})$ involves computing the hessian of the reinforcement learning objective, i.e., $\nabla_\theta^2 \mathbb{E}_{\boldsymbol{\tau}\sim P_\mathcal{T}(\boldsymbol{\tau}|\theta)}[R(\boldsymbol{\tau})]$. Estimating this hessian has been discussed in Baxter & Bartlett (2001) and Furmston et al. (2016). In the infinite horizon MDP case, Baxter & Bartlett (2001) derived a decomposition of the hessian. We extend their finding to the finite horizon case, showing that the hessian can be decomposed into three matrix terms (see Appendix B.2 for proof):

$$\nabla_\theta U(\theta, \mathcal{T}) = I + \alpha\nabla_\theta^2 \mathbb{E}_{\boldsymbol{\tau}\sim P_\mathcal{T}(\boldsymbol{\tau}|\theta)}[R(\boldsymbol{\tau})] = I + \alpha\big(\mathcal{H}_1 + \mathcal{H}_2 + \mathcal{H}_{12} + \mathcal{H}_{12}^\top\big) \tag{6}$$

whereby

$$\mathcal{H}_1 = \mathbb{E}_{\boldsymbol{\tau}\sim P_\mathcal{T}(\boldsymbol{\tau}|\theta)}\left[\sum_{t=0}^{H-1}\nabla_\theta\log\pi_\theta(\boldsymbol{a}_t, \boldsymbol{s}_t)\nabla_\theta\log\pi_\theta(\boldsymbol{a}_t, \boldsymbol{s}_t)^\top\left(\sum_{t'=t}^{H-1}r(\boldsymbol{s}_{t'}, \boldsymbol{a}_{t'})\right)\right]$$

$$\mathcal{H}_2 = \mathbb{E}_{\boldsymbol{\tau}\sim P_\mathcal{T}(\boldsymbol{\tau}|\theta)}\left[\sum_{t=0}^{H-1}\nabla_\theta^2\log\pi_\theta(\boldsymbol{a}_t, \boldsymbol{s}_t)\left(\sum_{t'=t}^{H-1}r(\boldsymbol{s}_{t'}, \boldsymbol{a}_{t'})\right)\right]$$

$$\mathcal{H}_{12} = \mathbb{E}_{\boldsymbol{\tau}\sim P_\mathcal{T}(\boldsymbol{\tau}|\theta)}\left[\sum_{t=0}^{H-1}\nabla_\theta\log\pi_\theta(\boldsymbol{a}_t, \boldsymbol{s}_t)\nabla_\theta Q_t^{\pi_\theta}(\boldsymbol{s}_t, \boldsymbol{a}_t)^\top\right]$$

---

[1]Note that MAML is theoretically sound, but does not attend to correctly estimating the meta-policy gradients. As consequence, the gradients in the corresponding implementation do not comply with the theory.

Here $Q_t^{\pi_\theta}(\boldsymbol{s}_t, \boldsymbol{a}_t) = \mathbb{E}_{\boldsymbol{\tau}^{t+1:H-1} \sim P_{\mathcal{T}}(\cdot|\theta)} \left[ \sum_{t'=t}^{H-1} r(\boldsymbol{s}_{t'}, \boldsymbol{a}_{t'}) | s_t, a_t \right]$ denotes the expected state-action value function under policy $\pi_\theta$ at time $t$.

Computing the expectation of the RL-objective is in general intractable. Typically, its gradients are computed with a Monte Carlo estimate based on the policy gradient theorem (Eq. 82). In practical implementations, such an estimate is obtained by automatically differentiating a surrogate objective (Schulman et al., 2015b). However, this results in a highly biased hessian estimate which just computes $\mathcal{H}_2$, entirely dropping the terms $\mathcal{H}_1$ and $\mathcal{H}_{12} + \mathcal{H}_{12}^\top$. In the notation of the previous section, it leads to neglecting the $\nabla_\theta J_{\text{pre}}$ term, ignoring the influence of the pre-update sampling distribution.

The issue can be overcome using the DiCE formulation, which allows to compute unbiased higher-order Monte Carlos estimates of arbitrary stochastic computation graphs (Foerster et al., 2018). The DiCE-RL objective can be rewritten as follows

$$J^{\text{DiCE}}(\boldsymbol{\tau}) = \sum_{t=0}^{H-1} \left( \prod_{t'=0}^{t} \frac{\pi_\theta(\boldsymbol{a}_{t'}|\boldsymbol{s}_{t'})}{\perp(\pi_\theta(\boldsymbol{a}_{t'}|\boldsymbol{s}_{t'}))} \right) r(\boldsymbol{s}_t, \boldsymbol{a}_t) \quad \boldsymbol{\tau} \sim P_{\mathcal{T}}(\boldsymbol{\tau}) \tag{7}$$

$$\mathbb{E}_{\boldsymbol{\tau} \sim P_{\mathcal{T}}(\boldsymbol{\tau}|\theta)} \left[ \nabla_\theta^2 J^{\text{DiCE}}(\boldsymbol{\tau}) \right] = \mathcal{H}_1 + \mathcal{H}_2 + \mathcal{H}_{12} + \mathcal{H}_{12}^\top \tag{8}$$

In that, $\perp$ denotes the "stop_gradient" operator, i.e., $\perp(f_\theta(x)) \to f_\theta(x)$ but $\nabla_\theta \perp(f_\theta(x)) \to 0$. The sequential dependence of $\pi_\theta(\boldsymbol{a}_t|\boldsymbol{s}_t)$ within the trajectory, manifesting itself through the product of importance weights in (7), results in high variance estimates of the hessian $\nabla_\theta^2 \mathbb{E}_{\boldsymbol{\tau} \sim P_{\mathcal{T}}(\boldsymbol{\tau}|\theta)} [R(\boldsymbol{\tau})]$. As noted by Furmston et al. (2016), $\mathcal{H}_{12}$ is particularly difficult to estimate, since it involves three nested sums along the trajectory. In section 7.2 we empirically show that the high variance estimates of the DiCE objective lead to noisy meta-policy gradients and poor learning performance.

To facilitate a sample efficient meta-learning, we introduce the low variance curvature (LVC) estimator:

$$J^{\text{LVC}}(\boldsymbol{\tau}) = \sum_{t=0}^{H-1} \frac{\pi_\theta(\boldsymbol{a}_t|\boldsymbol{s}_t)}{\perp(\pi_\theta(\boldsymbol{a}_t|\boldsymbol{s}_t))} \left( \sum_{t'=t}^{H-1} r(\boldsymbol{s}_{t'}, \boldsymbol{a}_{t'}) \right) \quad \boldsymbol{\tau} \sim P_{\mathcal{T}}(\boldsymbol{\tau}) \tag{9}$$

$$\mathbb{E}_{\boldsymbol{\tau} \sim P_{\mathcal{T}}(\boldsymbol{\tau}|\theta)} \left[ \nabla_\theta^2 J^{\text{LVC}}(\boldsymbol{\tau}) \right] = \mathcal{H}_1 + \mathcal{H}_2 \tag{10}$$

By removing the sequential dependence of $\pi_\theta(\boldsymbol{a}_t|\boldsymbol{s}_t)$ within trajectories, the hessian estimate neglects the term $\mathcal{H}_{12} + \mathcal{H}_{12}^\top$ which leads to a variance reduction, but makes the estimate biased. The choice of this objective function is motivated by findings in Furmston et al. (2016): under certain conditions the term $\mathcal{H}_{12} + \mathcal{H}_{12}^\top$ vanishes around local optima $\theta^*$, i.e., $\mathbb{E}_{\boldsymbol{\tau}}[\nabla_\theta^2 J^{\text{LVC}}] \to \mathbb{E}_{\boldsymbol{\tau}}[\nabla_\theta^2 J^{\text{DiCE}}]$ as $\theta \to \theta^*$. Hence, the bias of the LVC estimator becomes negligible close to local optima. The experiments in section 7.2 underpin the theoretical findings, showing that the low variance hessian estimates obtained through $J^{\text{LVC}}$ improve the sample-efficiency of meta-learning by a significant margin when compared to $J^{\text{DiCE}}$. We refer the interested reader to Appendix B for derivations and a more detailed discussion.

## 6 PROMP: PROXIMAL META-POLICY SEARCH

Building on the previous sections, we develop a novel meta-policy search method based on the low variance curvature objective which aims to solve the following optimization problem:

$$\max_\theta \quad \mathbb{E}_{\mathcal{T} \sim \rho(\mathcal{T})} \left[ \mathbb{E}_{\boldsymbol{\tau}' \sim P_{\mathcal{T}}(\boldsymbol{\tau}'|\theta')} [R(\boldsymbol{\tau}')] \right] \quad \text{with} \quad \theta' := \theta + \alpha \, \nabla_\theta \mathbb{E}_{\boldsymbol{\tau} \sim P_{\mathcal{T}}(\boldsymbol{\tau}|\theta)} \left[ J^{\text{LVC}}(\boldsymbol{\tau}) \right] \tag{11}$$

Prior work has optimized this objective using either vanilla policy gradient (VPG) or TRPO (Schulman et al., 2015a). TRPO holds the promise to be more data efficient and stable during the learning process when compared to VPG. However, it requires computing the Fisher information matrix (FIM). Estimating the FIM is particularly problematic in the meta-learning set up. The meta-policy gradients already involve second order derivatives; as a result, the time complexity of the FIM estimate is cubic in the number of policy parameters. Typically, the problem is circumvented using finite difference methods, which introduce further approximation errors.

The recently introduced PPO algorithm (Schulman et al., 2017) achieves comparable results to TRPO with the advantage of being a first order method. PPO uses a surrogate clipping objective which allows it to safely take multiple gradient steps without re-sampling trajectories.

$$J_{\mathcal{T}}^{\text{CLIP}}(\theta) = \mathbb{E}_{\boldsymbol{\tau} \sim P_{\mathcal{T}}(\boldsymbol{\tau}, \theta_o)} \left[ \sum_{t=0}^{H-1} \min \left( \frac{\pi_\theta(\boldsymbol{a}_t|\boldsymbol{s}_t)}{\pi_{\theta_o}(\boldsymbol{a}_t|\boldsymbol{s}_t)} A^{\pi_{\theta_o}}(\boldsymbol{s}_t, \boldsymbol{a}_t) , \, \text{clip}_{1-\epsilon}^{1+\epsilon} \left( \frac{\pi_\theta(\boldsymbol{a}_t|\boldsymbol{s}_t)}{\pi_{\theta_o}(\boldsymbol{a}_t|\boldsymbol{s}_t)} \right) A^{\pi_{\theta_o}}(\boldsymbol{s}_t, \boldsymbol{a}_t) \right) \right]$$

---

**Algorithm 1** Proximal Meta-Policy Search (ProMP)

---

**Require:** Task distribution $\rho$, step sizes $\alpha$, $\beta$, KL-penalty coefficient $\eta$, clipping range $\epsilon$
1: Randomly initialize $\theta$
2: **while** $\theta$ not converged **do**
3:     Sample batch of tasks $\mathcal{T}_i \sim \rho(\mathcal{T})$
4:     **for** step $n = 0, ..., N - 1$ **do**
5:         **if** $n = 0$ **then**
6:             Set $\theta_o \leftarrow \theta$
7:             **for all** $\mathcal{T}_i \sim \rho(\mathcal{T})$ **do**
8:                 Sample pre-update trajectories $\mathcal{D}_i = \{\tau_i\}$ from $\mathcal{T}_i$ using $\pi_\theta$
9:                 Compute adapted parameters $\theta'_{o,i} \leftarrow \theta + \alpha \, \nabla_\theta J_{\mathcal{T}_i}^{LR}(\theta)$ with $\mathcal{D}_i = \{\tau_i\}$
10:                 Sample post-update trajectories $\mathcal{D}'_i = \{\tau'_i\}$ from $\mathcal{T}_i$ using $\pi_{\theta'_{o,i}}$
11:     Update $\theta \leftarrow \theta + \beta \sum_{\mathcal{T}_i} \nabla_\theta J_{\mathcal{T}_i}^{\text{ProMP}}(\theta)$ using each $\mathcal{D}'_i = \{\tau'_i\}$

---

In case of Meta-RL, it does not suffice to just replace the post-update reward objective with $J_{\mathcal{T}}^{\text{CLIP}}$. In order to safely perform multiple meta-gradient steps based on the same sampled data from a recent policy $\pi_{\theta_o}$, we also need to 1) account for changes in the pre-update action distribution $\pi_\theta(a_t|s_t)$, and 2) bound changes in the pre-update state visitation distribution (Kakade & Langford, 2002).

We propose Proximal Meta-Policy Search (ProMP) which incorporates both the benefits of proximal policy optimization and the low variance curvature objective (see Alg. 1.) In order to comply with requirement 1), ProMP replaces the "stop gradient" importance weight $\frac{\pi_\theta(a_t|s_t)}{\perp(\pi_\theta(a_t|s_t))}$ by the likelihood ratio $\frac{\pi_\theta(a_t|s_t)}{\pi_{\theta_o}(a_t|s_t))}$, which results in the following objective

$$J_{\mathcal{T}}^{LR}(\theta) = \mathbb{E}_{\boldsymbol{\tau} \sim P_{\mathcal{T}}(\boldsymbol{\tau}, \theta_o)} \left[ \sum_{t=0}^{H-1} \frac{\pi_\theta(\boldsymbol{a}_t|\boldsymbol{s}_t)}{\pi_{\theta_o}(\boldsymbol{a}_t|\boldsymbol{s}_t)} A^{\pi_{\theta_o}}(\boldsymbol{s}_t, \boldsymbol{a}_t) \right] \tag{12}$$

An important feature of this objective is that its derivatives w.r.t $\theta$ evaluated at $\theta_o$ are identical to those of the LVC objective, and it additionally accounts for changes in the pre-update action distribution. To satisfy condition 2) we extend the clipped meta-objective with a KL-penalty term between $\pi_\theta$ and $\pi_{\theta_o}$. This KL-penalty term enforces a soft local "trust region" around $\pi_{\theta_o}$, preventing the shift in state visitation distribution to become large during optimization. This enables us to take multiple meta-policy gradient steps without re-sampling. Altogether, ProMP optimizes

$$J_{\mathcal{T}}^{\text{ProMP}}(\theta) = J_{\mathcal{T}}^{\text{CLIP}}(\theta') - \eta \bar{\mathcal{D}}_{KL}(\pi_{\theta_o}, \pi_\theta) \quad \text{s.t.} \quad \theta' = \theta + \alpha \, \nabla_\theta J_{\mathcal{T}}^{LR}(\theta) \,, \quad \mathcal{T} \sim \rho(\mathcal{T}) \tag{13}$$

ProMP consolidates the insights developed throughout the course of this paper, while at the same time making maximal use of recently developed policy gradients algorithms. First, its meta-learning formulation exploits the full structural knowledge of gradient-based meta-learning. Second, it incorporates a low variance estimate of the RL-objective hessian. Third, ProMP controls the statistical distance of both pre- and post-adaptation policies, promoting efficient and stable meta-learning. All in all, ProMP consistently outperforms previous gradient-based meta-RL algorithms in sample complexity, wall clock time, and asymptotic performance (see Section 7.1).

## 7 EXPERIMENTS

In order to empirically validate the theoretical arguments outlined above, this section provides a detailed experimental analysis that aims to answer the following questions: (i) How does ProMP perform against previous Meta-RL algorithms? (ii) How do the lower variance but biased LVC gradient estimates compare to the high variance, unbiased DiCE estimates? (iii) Do the different formulations result in different pre-update exploration properties? (iv) How do formulation I and formulation II differ in their meta-gradient estimates and convergence properties?

To answer the posed questions, we evaluate our approach on six continuous control Meta-RL benchmark environments based on OpenAI Gym and the Mujoco simulator (Brockman et al., 2016; Todorov et al., 2012). A description of the experimental setup is found in Appendix D. In all experiments, the reported curves are averaged over at least three random seeds. Returns are estimated

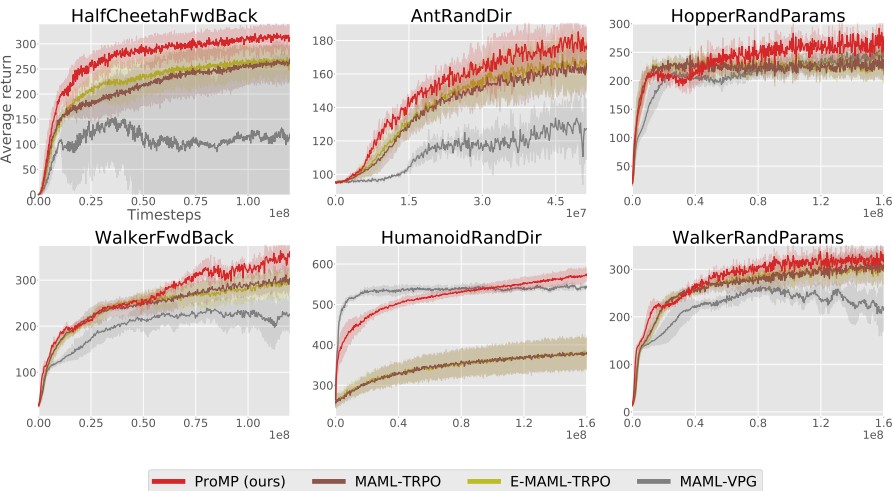

Figure 2: Meta-learning curves of ProMP and previous gradient-based meta-learning algorithms in six different MuJoCo environments. ProMP outperforms previous work in all the the environments.

based on sampled trajectories from the adapted post-update policies and averaged over sampled tasks. The source code and the experiment data are available on our supplementary website.[2]

## 7.1 META-GRADIENT BASED COMPARISON

We compare our method, ProMP, in sample complexity and asymptotic performance to the gradient-based meta-learning approaches MAML-TRPO (Finn et al., 2017) and E-MAML-TRPO (see Fig. 2). Note that MAML corresponds to the original implementation of RL-MAML by (Finn et al., 2017) where no credit assignment to the pre-adaptation policy is happening (see Appendix B.3 for details). Moreover, we provide a second study which focuses on the underlying meta-gradient estimator. Specifically, we compare the LVC, DiCE, MAML and E-MAML estimators while optimizing meta-learning objective with vanilla policy gradient (VPG) ascent. This can be viewed as an ablated version of the algorithms which tries to eliminate the influences of the outer optimizers on the learning performance (see Fig. 3).

These algorithms are benchmarked on six different locomotion tasks that require adaptation: the half-cheetah and walker must switch between running forward and backward, the high-dimensional agents ant and humanoid must learn to adapt to run in different directions in the 2D-plane, and the hopper and walker have to adapt to different configuration of their dynamics.

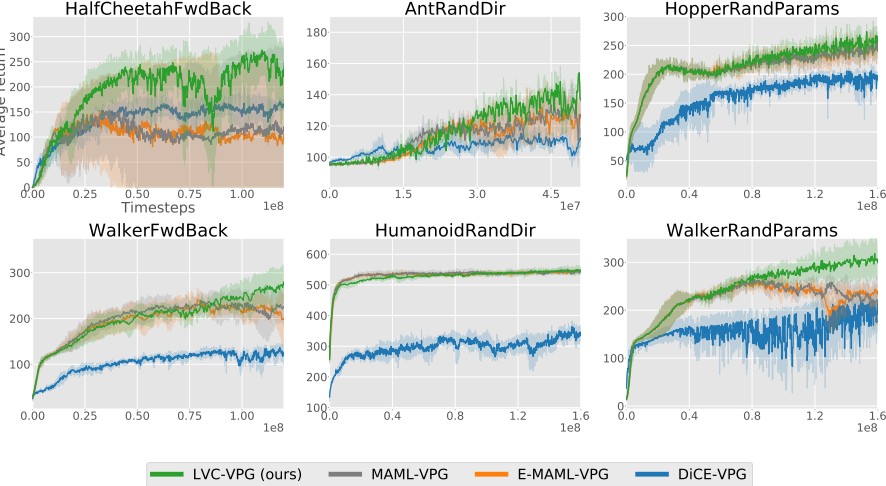

Figure 3: Meta-learning curves corresponding to different meta-gradient estimators in conjunction with VPG. The introduced LVC approach consistently outperforms the other estimators.

[2]https://sites.google.com/view/pro-mp

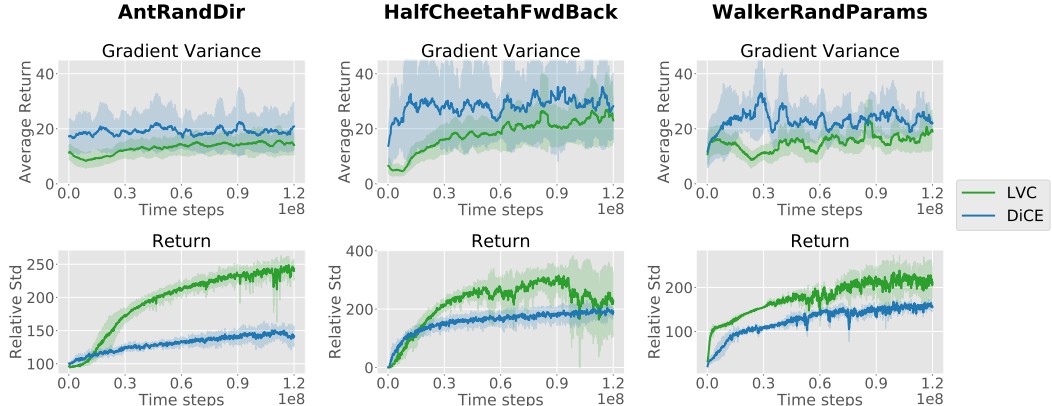

Figure 4: Top: Relative standard deviation of meta-policy gradients. Bottom: Returns in the respective environments throughout the learning process. LVC exhibits less variance in its meta-gradients which may explain its superior performance when compared to DiCE.

The results in Figure 2 highlight the strength of ProMP in terms of sample efficiency and asymptotic performance. In the meta-gradient estimator study in Fig. 3, we demonstrate the positive effect of the LVC objective, as it consistently outperforms the other estimators. In contrast, DiCE learns only slowly when compared to the other approaches. As we have motivated mathematically and substantiate empirically in the following experiment, the poor performance of DiCE may be ascribed to the high variance of its meta-gradient estimates. The fact that the results of MAML and E-MAML are comparable underpins the ineffectiveness of the naive pre-update credit assignment (i.e. formulation II), as discussed in section 4.

Results for four additional environments are displayed in Appendix D along with hyperparameter settings, environment specifications and a wall-clock time comparison of the algorithms.

### 7.2 Gradient Estimator Variance and Its Effect on Meta-Learning

In Section 5 we discussed how the DiCE formulation yields unbiased but high variance estimates of the RL-objective hessian and served as motivation for the low variance curvature (LVC) estimator. Here we investigate the meta-gradient variance of both estimators as well as its implication on the learning performance. Specifically, we report the relative standard deviation of the meta-policy gradients as well as the average return throughout the learning process in three of the meta-environments.

The results, depicted in Figure 4, highlight the advantage of the low variance curvature estimate. The trajectory level dependencies inherent in the DiCE estimator leads to a meta-gradient standard deviation that is on average 60% higher when compared to LVC. As the learning curves indicate, the noisy gradients may be a driving factor for the poor performance of DiCE, impeding sample efficient meta-learning. Meta-policy search based on the LVC estimator leads to substantially better sample-efficiency and asymptotic performance.

In case of HalfCheetahFwdBack, we observe some unstable learning behavior of LVC-VPG which is most likely caused by the bias of LVC in combination with the naive VPG optimizer. However, the mechanisms in ProMP that ensure proximity w.r.t. to the policys KL-divergence seem to counteract these instabilities during training, giving us a stable and efficient meta-learning algorithm.

### 7.3 Comparison of Initial Sampling Distributions

Here we evaluate the effect of the different objectives on the learned pre-update sampling distribution. We compare the low variance curvature (LVC) estimator with TRPO (LVC-TRPO) against MAML (Finn et al., 2017) and E-MAML-TRPO (Stadie et al., 2018) in a 2D environment on which the exploration behavior can be visualized. Each task of this environment corresponds to reaching a different corner location; however, the 2D agent only experiences reward when it is sufficiently close to the corner (translucent regions of Figure 5). Thus, to successfully identify the task, the agent must explore the different regions. We perform three inner adaptation steps on each task, allowing the agent to fully change its behavior from exploration to exploitation.

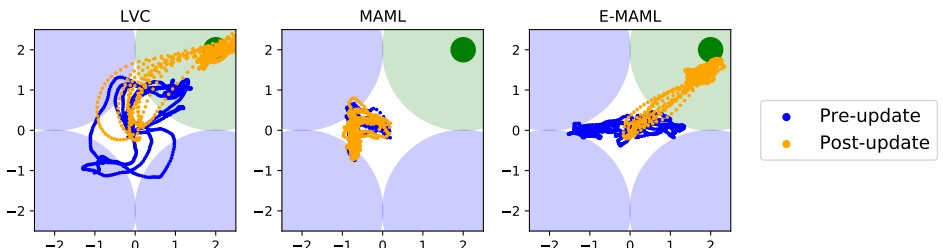

Figure 5: Exploration patterns of the pre-update policy and exploitation post-update with different update functions. Through its superior credit assignment, the LVC objective learns a pre-update policy that is able to identify the current task and respectively adapt its policy, successfully reaching the goal (dark green circle).

The different exploration-exploitation strategies are displayed in Figure 5. Since the MAML implementation does not assign credit to the pre-update sampling trajectory, it is unable to learn a sound exploration strategy for task identification and thus fails to accomplish the task. On the other hand, E-MAML, which corresponds to formulation II, learns to explore in long but random paths: because it can only assign credit to batches of pre-update trajectories, there is no notion of which actions in particular facilitate good task adaptation. As consequence the adapted policy slightly misses the task-specific target. The LVC estimator, instead, learns a consistent pattern of exploration, visiting each of the four regions, which it harnesses to fully solve the task.

## 7.4 GRADIENT UPDATE DIRECTIONS OF THE TWO META-RL FORMULATIONS

To shed more light on the differences of the gradients of formulation I and formulation II, we evaluate the meta-gradient updates and the corresponding convergence to the optimum of both formulations in a simple 1D environment. In this environment, the agent starts in a random position in the real line and has to reach a goal located at the position 1 or -1. In order to visualize the convergence, we parameterize the policy with only two parameters $\theta_0$ and $\theta_1$. We employ formulation I by optimizing the DiCE objective with VPG, and formulation II by optimizing its (E-MAML) objective with VPG.

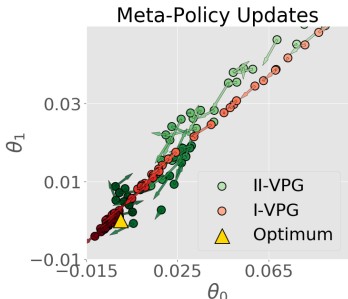

Figure 6 depicts meta-gradient updates of the parameters $\theta_i$ for both formulations. Formulation I (red) exploits the internal structure of the adaptation update yielding faster and steadier convergence to the optimum. Due to its inferior credit assignment, formulation II (green) produces noisier gradient estimates leading to worse convergence properties.

Figure 6: Meta-gradient updates of policy parameters $\theta_0$ and $\theta_1$ in a 1D environment w.r.t Formulation I (red) and Formulation II (green).

## 8 CONCLUSION

In this paper we propose a novel Meta-RL algorithm, proximal meta-policy search (ProMP), which fully optimizes for the pre-update sampling distribution leading to effective task identification. Our method is the result of a theoretical analysis of gradient-based Meta-RL formulations, based on which we develop the low variance curvature (LVC) surrogate objective that produces low variance meta-policy gradient estimates. Experimental results demonstrate that our approach surpasses previous meta-reinforcement learning approaches in a diverse set of continuous control tasks. Finally, we underpin our theoretical contributions with illustrative examples which further justify the soundness and effectiveness of our method.

## ACKNOWLEDGMENTS

Ignasi Clavera was supported by the La Caixa Fellowship. The research leading to these results has received funding from the German Research Foundation (DFG: Deutsche Forschungsgemeinschaft) under Priority Program on Autonomous Learning (SPP 1527) and was supported by Berkeley Deep Drive, Amazon Web Services, and Huawei. Also we thank Abhishek Gupta, Chelsea Finn, aand Aviv Tamar for their valuable feedback.

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

## A  TWO META-POLICY GRADIENT FORMULATIONS

In this section we discuss two different gradient-based meta-learning formulations, derive their gradients and analyze the differences between them.

### A.1  META-POLICY GRADIENT FORMULATION I

The first meta-learning formulation, known as MAML (Finn et al., 2017), views the inner update rule $U(\theta, \mathcal{T})$ as a mapping from the pre-update parameter $\theta$ and the task $\mathcal{T}$ to an adapted policy parameter $\theta'$. The update function can be viewed as stand-alone procedure that encapsulates sampling from the task-specific trajectory distribution $P_{\mathcal{T}}(\boldsymbol{\tau}|\pi_\theta)$ and updating the policy parameters. Building on this concept, the meta-objective can be written as

$$J^I(\theta) = \mathbb{E}_{\mathcal{T}\sim\rho(\mathcal{T})}\left[\mathbb{E}_{\boldsymbol{\tau}'\sim P_{\mathcal{T}}(\boldsymbol{\tau}'|\theta')}[R(\boldsymbol{\tau}')]\right] \quad \text{with} \quad \theta' := U(\theta, \mathcal{T}) \tag{14}$$

The task-specific gradients follow as

$$\nabla_\theta J^I_{\mathcal{T}}(\theta) = \nabla_\theta E_{\boldsymbol{\tau}'\sim P_{\mathcal{T}}(\boldsymbol{\tau}'|\theta')}[R(\boldsymbol{\tau}')] \tag{15}$$

$$= E_{\boldsymbol{\tau}'\sim P_{\mathcal{T}}(\boldsymbol{\tau}'|\theta')}[\nabla_\theta \log P_{\mathcal{T}}(\boldsymbol{\tau}'|\theta')R(\boldsymbol{\tau}')] \tag{16}$$

$$= E_{\boldsymbol{\tau}'\sim P_{\mathcal{T}}(\boldsymbol{\tau}'|\theta')}[\nabla_{\theta'} \log P_{\mathcal{T}}(\boldsymbol{\tau}'|\theta')R(\boldsymbol{\tau}')\nabla_\theta\theta'] \tag{17}$$

In order to derive the gradients of the inner update $\nabla_\theta\theta' = \nabla_\theta U(\theta, \mathcal{T})$ it is necessary to know the structure of $U$. The main part of this paper assumes the inner update rule to be a policy gradient descent step

$$\nabla_\theta U(\theta, \mathcal{T}) = \nabla_\theta\left(\theta + \alpha\,\nabla_\theta \mathbb{E}_{\boldsymbol{\tau}\sim P_{\mathcal{T}}(\boldsymbol{\tau}|\theta)}[R(\boldsymbol{\tau})]\right) \tag{18}$$

$$= I + \alpha\nabla_\theta^2\,\mathbb{E}_{\boldsymbol{\tau}\sim P_{\mathcal{T}}(\boldsymbol{\tau}|\theta)}[R(\boldsymbol{\tau})] \tag{19}$$

Thereby the second term in (19) is the local curvature (hessian) of the inner adaptation objective function. The correct hessian of the inner objective can be derived as follows:

$$\nabla_\theta^2\,\mathbb{E}_{\boldsymbol{\tau}\sim P_{\mathcal{T}}(\boldsymbol{\tau}|\theta)}[R(\boldsymbol{\tau})] = \nabla_\theta\,\mathbb{E}_{\boldsymbol{\tau}\sim P_{\mathcal{T}}(\boldsymbol{\tau}|\theta)}[\nabla_\theta \log \pi_\theta(\boldsymbol{\tau})R(\boldsymbol{\tau})] \tag{20}$$

$$= \nabla_\theta\,\int P_{\mathcal{T}}(\boldsymbol{\tau}|\theta)\nabla_\theta \log \pi_\theta(\boldsymbol{\tau})R(\boldsymbol{\tau})d\boldsymbol{\tau} \tag{21}$$

$$= \int P_{\mathcal{T}}(\boldsymbol{\tau}|\theta)\nabla_\theta \log \pi_\theta(\boldsymbol{\tau})\nabla_\theta \log \pi_\theta(\boldsymbol{\tau})^\top R(\boldsymbol{\tau})+ \tag{22}$$

$$P_{\mathcal{T}}(\boldsymbol{\tau}|\theta)\nabla_\theta^2 \log \pi_\theta(\boldsymbol{\tau})R(\boldsymbol{\tau})d\boldsymbol{\tau} \tag{23}$$

$$= \mathbb{E}_{\boldsymbol{\tau}\sim P_{\mathcal{T}}(\boldsymbol{\tau}|\theta)}\left[R(\boldsymbol{\tau})\left(\nabla_\theta^2 \log \pi_\theta(\boldsymbol{\tau}) + \nabla_\theta \log \pi_\theta(\boldsymbol{\tau})\nabla_\theta \log \pi_\theta(\boldsymbol{\tau})^\top\right)\right] \tag{24}$$

### A.2  META-POLICY GRADIENT FORMULATION II

The second meta-reinforcement learning formulation views the the inner update $\theta' = U(\theta, \tau^{1:N})$ as a deterministic function of the pre-update policy parameters $\theta$ and $N$ trajectories $\boldsymbol{\tau}^{1:N} \sim P_{\mathcal{T}}(\boldsymbol{\tau}^{1:N}|\theta)$ sampled from the pre-update trajectory distribution. This formulation was introduced in Al-Shedivat et al. (2018) and further discussed with respect to its exploration properties in Stadie et al. (2018).

Viewing $U$ as a function that adapts the policy parameters $\theta$ to a specific task $\mathcal{T}$ given policy rollouts in this task, the corresponding meta-learning objective can be written as

$$J^{II}(\theta) = \mathbb{E}_{\mathcal{T}\sim\rho(\mathcal{T})}\left[\mathbb{E}_{\boldsymbol{\tau}^{1:N}\sim P_{\mathcal{T}}(\boldsymbol{\tau}^{1:N}|\theta)}\left[\mathbb{E}_{\boldsymbol{\tau}'\sim P_{\mathcal{T}}(\boldsymbol{\tau}'|\theta')}[R(\boldsymbol{\tau}')]\right]\right] \quad \text{with} \quad \theta' := U(\theta, \boldsymbol{\tau}^{1:N}) \tag{25}$$

Since the first part of the gradient derivation is agnostic to the inner update rule $U(\theta, \tau^{1:N})$, we only assume that the inner update function $U$ is differentiable w.r.t. $\theta$. First we rewrite the meta-objective $J(\theta)$ as expectation of task specific objectives $J^{II}_{\mathcal{T}}(\theta)$ under the task distribution. This allows us to express the meta-policy gradients as expectation of task-specific gradients:

$$\nabla_\theta J^{II}(\theta) = \mathbb{E}_{\mathcal{T}\sim\rho(\mathcal{T})}\left[\nabla_\theta J^{II}_{\mathcal{T}}(\theta)\right] \tag{26}$$

The task specific gradients can be calculated as follows

$$
\begin{aligned}
\nabla_\theta J_\mathcal{T}^{II}(\theta) =& \ \nabla_\theta \mathbb{E}_{\boldsymbol{\tau} \sim P_\mathcal{T}(\boldsymbol{\tau}^{1:N}|\theta)} \Big[ \mathbb{E}_{\boldsymbol{\tau}' \sim P_\mathcal{T}(\boldsymbol{\tau}'|\theta')} \big[ R(\boldsymbol{\tau}') \big] \Big] \\
=& \ \nabla_\theta \int \int R(\boldsymbol{\tau}') \, P_\mathcal{T}(\boldsymbol{\tau}'|\theta') \, P_\mathcal{T}(\boldsymbol{\tau}^{1:N}|\theta) \, d\boldsymbol{\tau}' \, d\boldsymbol{\tau} \\
=& \ \int \int R(\boldsymbol{\tau}') \, P_\mathcal{T}(\boldsymbol{\tau}'|\theta') \, \nabla_\theta \log P_\mathcal{T}(\boldsymbol{\tau}^{1:N}|\theta) P_\mathcal{T}(\boldsymbol{\tau}^{1:N}|\theta) + \\
& \qquad R(\boldsymbol{\tau}') \, \nabla_\theta \log P_\mathcal{T}(\boldsymbol{\tau}'|\theta') P_\mathcal{T}(\boldsymbol{\tau}'|\theta') \, P_\mathcal{T}(\boldsymbol{\tau}^{1:N}|\theta) \, d\boldsymbol{\tau}' \, d\boldsymbol{\tau} \\
=& \ \mathbb{E}_{\substack{\boldsymbol{\tau}^{1:N} \sim P_\mathcal{T}(\boldsymbol{\tau}^{1:N}|\theta) \\ \boldsymbol{\tau}' \sim P_\mathcal{T}(\boldsymbol{\tau}'|\theta')}} \Bigg[ R(\boldsymbol{\tau}') \bigg( \nabla_\theta \log P_\mathcal{T}(\boldsymbol{\tau}'|\theta') + \sum_{i=1}^N \nabla_\theta \log P_\mathcal{T}(\boldsymbol{\tau}^{(n)}|\theta) \bigg) \Bigg] \\
=& \ \mathbb{E}_{\substack{\boldsymbol{\tau}^{1:N} \sim P_\mathcal{T}(\boldsymbol{\tau}^{1:N}|\theta) \\ \boldsymbol{\tau}' \sim P_\mathcal{T}(\boldsymbol{\tau}'|\theta')}} \Bigg[ R(\boldsymbol{\tau}') \bigg( \nabla_{\theta'} \log P_\mathcal{T}(\boldsymbol{\tau}'|\theta') \nabla_\theta \theta' + \sum_{n=1}^N \nabla_\theta \log P_\mathcal{T}(\boldsymbol{\tau}^{(n)}|\theta) \bigg) \Bigg]
\end{aligned}
$$

As in A.1 the structure of $U(\theta, \boldsymbol{\tau}^{1:N})$ must be known in order to derive the gradient $\nabla_\theta \theta'$. Since we assume the inner update to be vanilla policy gradient, the respective gradient follows as

$$
U(\theta, \tau^{1:N}) = \theta + \alpha \frac{1}{N} \sum_{n=1}^N \nabla_\theta \log \pi_\theta(\tau^{(n)})) R(\tau^{(n)}) \quad \text{with} \quad \nabla_\theta \log \pi_\theta(\tau) = \sum_{t=0}^{H-1} \nabla_\theta \log \pi_\theta(a_t|s_t)
$$

The respective gradient of $U(\theta, \tau^{1:N})$ follows as

$$
\nabla_\theta U(\theta, \tau^{1:N}) = \nabla_\theta \left( \theta + \alpha \frac{1}{N} \sum_{n=1}^N \nabla_\theta \log \pi_\theta(\tau^{(n)})) R(\tau^{(n)}) \right) \tag{27}
$$

$$
= I + \alpha \frac{1}{N} \sum_{n=1}^N \nabla_\theta^2 \log \pi_\theta(\tau^{(n)})) R(\tau^{(n)}) \tag{28}
$$

### A.3 COMPARING THE GRADIENTS OF THE TWO FORMULATIONS

In the following we analyze the differences between the gradients derived for the two formulations. To do so, we begin with $\nabla_\theta J_\mathcal{T}^I(\theta)$ by inserting the gradient of the inner adaptation step (19) into (17):

$$
\nabla_\theta J_\mathcal{T}^I(\theta) = E_{\boldsymbol{\tau}' \sim P_\mathcal{T}(\boldsymbol{\tau}'|\theta')} \left[ \nabla_{\theta'} \log P_\mathcal{T}(\boldsymbol{\tau}'|\theta') R(\boldsymbol{\tau}') \left( I + \alpha \nabla_\theta^2 \, \mathbb{E}_{\boldsymbol{\tau} \sim P_\mathcal{T}(\boldsymbol{\tau}|\theta)} \left[ R(\boldsymbol{\tau}) \right] \right) \right] \tag{29}
$$

We can substitute the hessian of the inner objective by its derived expression from (24) and then rearrange the terms. Also note that $\nabla_\theta \log P_\mathcal{T}(\boldsymbol{\tau}|\theta) = \nabla_\theta \log \pi_\theta(\boldsymbol{\tau}) = \sum_{t=1}^{H-1} \log \pi_\theta(\boldsymbol{a}_t|\boldsymbol{s}_t)$ where $H$ is the MDP horizon.

$$
\nabla_\theta J_\mathcal{T}^I(\theta) = E_{\boldsymbol{\tau}' \sim P_\mathcal{T}(\boldsymbol{\tau}'|\theta')} \Bigg[ \nabla_{\theta'} \log P_\mathcal{T}(\boldsymbol{\tau}'|\theta') R(\boldsymbol{\tau}') \bigg( I + \alpha \mathbb{E}_{\boldsymbol{\tau} \sim P_\mathcal{T}(\boldsymbol{\tau}|\theta)} \big[ R(\boldsymbol{\tau}) \tag{30}
$$

$$
\big( \nabla_\theta^2 \log \pi_\theta(\boldsymbol{\tau}) + \nabla_\theta \log \pi_\theta(\boldsymbol{\tau}) \nabla_\theta \log \pi_\theta(\boldsymbol{\tau})^\top \big) \big] \bigg) \Bigg] \tag{31}
$$

$$
= \mathbb{E}_{\substack{\boldsymbol{\tau} \sim P_\mathcal{T}(\boldsymbol{\tau}|\theta) \\ \boldsymbol{\tau}' \sim P_\mathcal{T}(\boldsymbol{\tau}'|\theta')}} \Bigg[ \underbrace{\nabla_{\theta'} \log \pi_{\theta'}(\boldsymbol{\tau}') R(\boldsymbol{\tau}') \Big( I + \alpha R(\boldsymbol{\tau}) \nabla_\theta^2 \log \pi_\theta(\boldsymbol{\tau}) \Big)}_{\nabla_\theta J_{\text{post}}(\boldsymbol{\tau}, \boldsymbol{\tau}')} \tag{32}
$$

$$
\underbrace{+ \alpha \nabla_{\theta'} \log \pi_{\theta'}(\boldsymbol{\tau}') R(\boldsymbol{\tau}') R(\boldsymbol{\tau}) \nabla_\theta \log \pi_\theta(\boldsymbol{\tau}) \nabla_\theta \log \pi_\theta(\boldsymbol{\tau})^\top}_{\nabla_\theta J_{\text{pre}}^I(\boldsymbol{\tau}, \boldsymbol{\tau}')} \Bigg] \tag{33}
$$

Next, we rearrange the gradient of $J^{II}$ into a similar form as $\nabla_\theta J^I_{\mathcal{T}}(\theta)$. For that, we start by inserting (28) for $\nabla_\theta \theta'$ and replacing the expectation over pre-update trajectories $\boldsymbol{\tau}^{1:N}$ by the expectation over a single trajectory $\boldsymbol{\tau}$.

$$\nabla_\theta J^I_{\mathcal{T}}(\theta) = \mathbb{E}_{\substack{\boldsymbol{\tau}\sim P_{\mathcal{T}}(\boldsymbol{\tau}|\theta) \\ \boldsymbol{\tau}'\sim P_{\mathcal{T}}(\boldsymbol{\tau}'|\theta')}} \left[ \underbrace{R(\boldsymbol{\tau}')\nabla_{\theta'}\log \pi_\theta(\boldsymbol{\tau}')\left(I + \alpha R(\boldsymbol{\tau})\nabla^2_\theta \log \pi_\theta(\tau)\right)}_{\nabla_\theta J_{\text{post}}(\boldsymbol{\tau},\boldsymbol{\tau}')} \right. \tag{34}$$

$$\left. \underbrace{+ R(\boldsymbol{\tau}')\nabla_\theta \log \pi_\theta(\boldsymbol{\tau})}_{\nabla_\theta J^I_{\text{pre}}(\boldsymbol{\tau},\boldsymbol{\tau}')} \right] \tag{35}$$

While the first part of the gradients match ((32) and (34)), the second part ((33) and (35)) differs. Since the second gradient term can be viewed as responsible for shifting the pre-update sampling distribution $P_{\mathcal{T}}(\boldsymbol{\tau}|\theta)$ towards higher post-update returns, we refer to it as $\nabla_\theta J_{\text{pre}}(\boldsymbol{\tau},\boldsymbol{\tau}')$ . To further analyze the difference between $\nabla_\theta J^I_{\text{pre}}$ and $\nabla_\theta J^{II}_{\text{pre}}$ we slightly rearrange (33) and put both gradient terms next to each other:

$$\nabla_\theta J^I_{\text{pre}}(\boldsymbol{\tau},\boldsymbol{\tau}') = \alpha \nabla_\theta \log \pi_\theta(\boldsymbol{\tau}) \left( \underbrace{(\nabla_\theta \log \pi_\theta(\boldsymbol{\tau})R(\boldsymbol{\tau}))^\top}_{\nabla_\theta J^{\text{inner}}} \underbrace{(\nabla_{\theta'}\log \pi_{\theta'}(\boldsymbol{\tau}')R(\boldsymbol{\tau}'))}_{\nabla_{\theta'} J^{\text{outer}}} \right) \tag{36}$$

$$\nabla_\theta J^{II}_{\text{pre}}(\boldsymbol{\tau},\boldsymbol{\tau}') = \alpha \nabla_\theta \log \pi_\theta(\boldsymbol{\tau})R(\boldsymbol{\tau}') \tag{37}$$

In the following we interpret and and compare of the derived gradient terms, aiming to provide intuition for the differences between the formulations:

The first gradient term $J_{\text{post}}$ that matches in both formulations corresponds to a policy gradient step on the post-update policy $\pi_{\theta'}$. Since $\theta'$ itself is a function of $\theta$, the term $\left(I + \alpha R(\boldsymbol{\tau})\nabla^2_\theta \log \pi_\theta(\tau)\right)$ can be seen as linear transformation of the policy gradient update $R(\boldsymbol{\tau}')\nabla_{\theta'}\log \pi_\theta(\boldsymbol{\tau}')$ from the post-update parameter $\theta'$ into $\theta$. Although $J_{\text{post}}$ takes into account the functional relationship between $\theta'$ and $\theta$, it does not take into account the pre-update sampling distribution $P_{\mathcal{T}}(\boldsymbol{\tau}|\theta)$.

This is where $\nabla_\theta J_{\text{pre}}$ comes into play: $\nabla_\theta J^I_{\text{pre}}$ can be viewed as policy gradient update of the pre-update policy $\pi_\theta$ w.r.t. to the post-update return $R(\tau')$. Hence this gradient term aims to shift the pre-update sampling distribution so that higher post-update returns are achieved. However, $\nabla_\theta J^{II}_{\text{pre}}$ does not take into account the causal dependence of the post-update policy on the pre-update policy. Thus a change in $\theta$ due to $\nabla_\theta J^{II}_{\text{pre}}$ may counteract the change due to $\nabla_\theta J^{II}_{\text{post}}$. In contrast, $\nabla_\theta J^I_{\text{pre}}$ takes the dependence of the the post-update policy on the pre-update sampling distribution into account. Instead of simply weighting the gradients of the pre-update policy $\nabla_\theta \log \pi_\theta(\boldsymbol{\tau})$ with $R(\tau')$ as in $\nabla_\theta J^I_{\text{post}}$, $\nabla_\theta J^I_{\text{post}}$ weights the gradients with inner product of the pre-update and post-update policy gradients. This inner product can be written as

$$\nabla_\theta J^{\text{inner}\top} \nabla_{\theta'} J^{\text{outer}} = ||\nabla_\theta J^{\text{inner}}||_2 \cdot ||\nabla_{\theta'} J^{\text{outer}}||_2 \cdot \cos(\delta) \tag{38}$$

wherein $\delta$ denotes the angle between the the inner and outer pre-update and post-update policy gradients. Hence, $\nabla_\theta J^I_{\text{post}}$ steers the pre-update policy towards not only towards larger post-updates returns but also towards larger adaptation steps $\alpha \nabla_\theta J^{\text{inner}}$, and better alignment of pre- and post-update policy gradients. This directly optimizes for maximal improvement / adaptation for the respective task. See Li et al. (2017); Nichol et al. (2018) for a comparable analysis in case of domain generalization and supervised meta-learning. Also note that (38) allows formulation I to perform credit assignment on the trajectory level whereas formulation II can only assign credit to entire batches of $N$ pre-update trajectories $\tau^{1:N}$.

As a result, we expect the first meta-policy gradient formulation to learn faster and more stably since the respective gradients take the dependence of the pre-update returns on the pre-update sampling distribution into account while this causal link is neglected in the second formulation.

# B    ESTIMATING THE META-POLICY GRADIENTS

When employing formulation I for gradient-based meta-learning, we aim maximize the loss

$$J(\theta) = \mathbb{E}_{\mathcal{T}\sim\rho(\mathcal{T})} \left[\mathbb{E}_{\boldsymbol{\tau}'\sim P_{\mathcal{T}}(\boldsymbol{\tau}'|\theta')}\left[R(\boldsymbol{\tau}')\right]\right] \quad \text{with} \quad \theta' := \theta + \alpha\,\nabla_\theta \mathbb{E}_{\boldsymbol{\tau}\sim P_{\mathcal{T}}(\boldsymbol{\tau}|\theta)}\left[R(\boldsymbol{\tau})\right] \tag{39}$$

by performing a form of gradient-descent on $J(\theta)$. Note that we, from now on, assume $J := J^I$ and thus omit the superscript indicating the respective meta-learning formulation. As shown in A.2 the gradient can be derived as $\nabla_\theta J(\theta) = \mathbb{E}_{(T) \sim \rho(T)}[\nabla_\theta J_{\mathcal{T}}(\theta)]$ with

$$\nabla_\theta J_{\mathcal{T}}(\theta) = E_{\boldsymbol{\tau}' \sim P_{\mathcal{T}}(\boldsymbol{\tau}'|\theta')}\left[\nabla_{\theta'} \log P_{\mathcal{T}}(\boldsymbol{\tau}'|\theta')R(\boldsymbol{\tau}')\left(I + \alpha \nabla_\theta^2 \mathbb{E}_{\boldsymbol{\tau} \sim P_{\mathcal{T}}(\boldsymbol{\tau}|\theta)}[R(\boldsymbol{\tau})]\right)\right] \quad (40)$$

where $\nabla_\theta^2 J_{\text{inner}}(\theta) := \nabla_\theta^2 \mathbb{E}_{\boldsymbol{\tau} \sim P_{\mathcal{T}}(\boldsymbol{\tau}|\theta)}[R(\boldsymbol{\tau})]$ denotes hessian of the inner adaptation objective w.r.t. $\theta$. This section concerns the question of how to properly estimate this hessian.

## B.1 Estimating Gradients of the RL Reward Objective

Since the expectation over the trajectory distribution $P_{\mathcal{T}}(\boldsymbol{\tau}|\theta)$ is in general intractable, the score function trick is typically used to used to produce a Monte Carlo estimate of the policy gradients. Although the gradient estimate can be directly defined, when using a automatic-differentiation toolbox it is usually more convenient to use an objective function whose gradients correspond to the policy gradient estimate. Due to the Policy Gradient Theorem (PGT) Sutton et al. (2000) such a "surrogate" objective can be written as:

$$\hat{J}^{\text{PGT}} = \frac{1}{K} \sum_{\tau_k} \sum_{t=0}^{H-1} \log \pi_\theta(a_t|s_t)\left(\sum_{t'=t}^{H} r(s_{t'}, a_{t'})\right) \quad \tau_k \sim P_{\mathcal{T}}(\tau) \quad (41)$$

$$= \frac{1}{K} \sum_{\tau_k} \sum_{t=0}^{H-1}\left(\sum_{t'=0}^{t} \log \pi_\theta(a_t|s_t)\right) r(s_{t'}, a_{t'}) \quad \tau_k \sim P_{\mathcal{T}}(\tau) \quad (42)$$

While (41) and (42) are equivalent (Peters & Schaal, 2006), the more popular formulation formulation (41) can be seen as forward looking credit assignment while (42) can be interpreted as backward looking credit assignment (Foerster et al., 2018). A generalized procedure for constructing "surrogate" objectives for arbitrary stochastic computation graphs can be found in Schulman et al. (2015a).

## B.2 A decomposition of the hessian

Estimating the the hessian of the reinforcement learning objective has been discussed in Furmston et al. (2016) and Baxter & Bartlett (2001) with focus on second order policy gradient methods. In the infinite horizon MDP case, Baxter & Bartlett (2001) derive a decomposition of the hessian. In the following, we extend their finding to the finite horizon case.

**Proposition.** The hessian of the RL objective can be decomposed into four matrix terms:

$$\nabla_\theta^2 J_{\text{inner}}(\theta) = \mathcal{H}_1 + \mathcal{H}_2 + \mathcal{H}_{12} + \mathcal{H}_{12}^\top \quad (43)$$

where

$$\mathcal{H}_1 = \mathbb{E}_{\boldsymbol{\tau} \sim P_{\mathcal{T}}(\boldsymbol{\tau}|\theta)}\left[\sum_{t=0}^{H-1} \nabla_\theta \log \pi_\theta(\boldsymbol{a}_t, \boldsymbol{s}_t)\nabla_\theta \log \pi_\theta(\boldsymbol{a}_t, \boldsymbol{s}_t)^\top\left(\sum_{t'=t}^{H-1} r(\boldsymbol{s}_{t'}, \boldsymbol{a}_{t'})\right)\right] \quad (44)$$

$$\mathcal{H}_2 = \mathbb{E}_{\boldsymbol{\tau} \sim P_{\mathcal{T}}(\boldsymbol{\tau}|\theta)}\left[\sum_{t=0}^{H-1} \nabla_\theta^2 \log \pi_\theta(\boldsymbol{a}_t, \boldsymbol{s}_t)\left(\sum_{t'=t}^{H-1} r(\boldsymbol{s}_{t'}, \boldsymbol{a}_{t'})\right)\right] \quad (45)$$

$$\mathcal{H}_{12} = \mathbb{E}_{\boldsymbol{\tau} \sim P_{\mathcal{T}}(\boldsymbol{\tau}|\theta)}\left[\sum_{t=0}^{H-1} \nabla_\theta \log \pi_\theta(\boldsymbol{a}_t, \boldsymbol{s}_t)\nabla_\theta Q_t^{\pi_\theta}(\boldsymbol{s}_t, \boldsymbol{a}_t)^\top\right] \quad (46)$$

Here $Q_t^{\pi_\theta}(\boldsymbol{s}_t, \boldsymbol{a}_t) = \mathbb{E}_{\boldsymbol{\tau}^{t+1:H-1} \sim P_{\mathcal{T}}(\cdot|\theta)}\left[\sum_{t'=t}^{H-1} r(\boldsymbol{s}_{t'}, \boldsymbol{a}_{t'})|s_t, a_t\right]$ denotes the expected state-action value function under policy $\pi_\theta$ at time $t$.

**Proof.** As derived in (24), the hessian of $J_{\text{inner}}(\theta)$ follows as:

$$\nabla_\theta^2 J_{\text{inner}} = \mathbb{E}_{\boldsymbol{\tau} \sim P_{\mathcal{T}}(\boldsymbol{\tau}|\theta)} \left[ R(\boldsymbol{\tau}) \left( \nabla_\theta^2 \log \pi_\theta(\boldsymbol{\tau}) + \nabla_\theta \log \pi_\theta(\boldsymbol{\tau}) \nabla_\theta \log \pi_\theta(\boldsymbol{\tau})^\top \right) \right] \tag{47}$$

$$= \mathbb{E}_{\boldsymbol{\tau} \sim P_{\mathcal{T}}(\boldsymbol{\tau}|\theta)} \left[ \sum_{t=0}^{H-1} \left( \sum_{t'=0}^{t} \nabla_\theta^2 \log \pi_\theta(\boldsymbol{a}_{t'}, \boldsymbol{s}_{t'}) \right) r(\boldsymbol{s}_t, \boldsymbol{a}_t) \right] \tag{48}$$

$$+ \mathbb{E}_{\boldsymbol{\tau} \sim P_{\mathcal{T}}(\boldsymbol{\tau}|\theta)} \left[ \sum_{t=0}^{H-1} \left( \sum_{t'=0}^{t} \nabla_\theta \log \pi_\theta(\boldsymbol{a}_{t'}, \boldsymbol{s}_{t'}) \right) \left( \sum_{t'=0}^{t} \nabla_\theta \log \pi_\theta(\boldsymbol{a}_{t'}, \boldsymbol{s}_{t'}) \right)^\top r(\boldsymbol{s}_t, \boldsymbol{a}_t) \right] \tag{49}$$

$$= \mathbb{E}_{\boldsymbol{\tau} \sim P_{\mathcal{T}}(\boldsymbol{\tau}|\theta)} \left[ \sum_{t=0}^{H-1} \nabla_\theta^2 \log \pi_\theta(\boldsymbol{a}_t, \boldsymbol{s}_t) \left( \sum_{t'=t}^{H-1} r(\boldsymbol{s}_{t'}, \boldsymbol{a}_{t'}) \right) \right] \tag{50}$$

$$+ \mathbb{E}_{\boldsymbol{\tau} \sim P_{\mathcal{T}}(\boldsymbol{\tau}|\theta)} \left[ \sum_{t=0}^{H-1} \left( \sum_{t'=0}^{t} \sum_{h=0}^{t} \nabla_\theta \log \pi_\theta(\boldsymbol{a}_{t'}, \boldsymbol{s}_{t'}) \nabla_\theta \log \pi_\theta(\boldsymbol{a}_h, \boldsymbol{s}_h)^\top \right) r(\boldsymbol{s}_t, \boldsymbol{a}_t) \right] \tag{51}$$

The term in (50) is equal to $\mathcal{H}_2$. We continue by showing that the remaining term in (51) is equivalent to $\mathcal{H}_1 + \mathcal{H}_{12} + \mathcal{H}_{12}^\top$. For that, we split the inner double sum in (51) into three components:

$$\mathbb{E}_{\boldsymbol{\tau} \sim P_{\mathcal{T}}(\boldsymbol{\tau}|\theta)} \left[ \sum_{t=0}^{H-1} \left( \sum_{t'=0}^{t} \sum_{h=0}^{t} \nabla_\theta \log \pi_\theta(\boldsymbol{a}_{t'}, \boldsymbol{s}_{t'}) \nabla_\theta \log \pi_\theta(\boldsymbol{a}_h, \boldsymbol{s}_h)^\top \right) r(\boldsymbol{s}_t, \boldsymbol{a}_t) \right] \tag{52}$$

$$= \mathbb{E}_{\boldsymbol{\tau} \sim P_{\mathcal{T}}(\boldsymbol{\tau}|\theta)} \left[ \sum_{t=0}^{H-1} \left( \sum_{t'=0}^{t} \nabla_\theta \log \pi_\theta(\boldsymbol{a}_{t'}, \boldsymbol{s}_{t'}) \nabla_\theta \log \pi_\theta(\boldsymbol{a}_{t'}, \boldsymbol{s}_{t'})^\top \right) r(\boldsymbol{s}_t, \boldsymbol{a}_t) \right] \tag{53}$$

$$+ \mathbb{E}_{\boldsymbol{\tau} \sim P_{\mathcal{T}}(\boldsymbol{\tau}|\theta)} \left[ \sum_{t=0}^{H-1} \left( \sum_{t'=0}^{t} \sum_{h=0}^{t'-1} \nabla_\theta \log \pi_\theta(\boldsymbol{a}_{t'}, \boldsymbol{s}_{t'}) \nabla_\theta \log \pi_\theta(\boldsymbol{a}_h, \boldsymbol{s}_h)^\top \right) r(\boldsymbol{s}_t, \boldsymbol{a}_t) \right] \tag{54}$$

$$+ \mathbb{E}_{\boldsymbol{\tau} \sim P_{\mathcal{T}}(\boldsymbol{\tau}|\theta)} \left[ \sum_{t=0}^{H-1} \left( \sum_{t'=0}^{t} \sum_{h=t'+1}^{t} \nabla_\theta \log \pi_\theta(\boldsymbol{a}_{t'}, \boldsymbol{s}_{t'}) \nabla_\theta \log \pi_\theta(\boldsymbol{a}_h, \boldsymbol{s}_h)^\top \right) r(\boldsymbol{s}_t, \boldsymbol{a}_t) \right] \tag{55}$$

By changing the backward looking summation over outer products into a forward looking summation of rewards, (53) can be shown to be equal to $\mathcal{H}_1$:

$$\mathbb{E}_{\boldsymbol{\tau} \sim P_{\mathcal{T}}(\boldsymbol{\tau}|\theta)} \left[ \sum_{t=0}^{H-1} \left( \sum_{t'=0}^{t} \nabla_\theta \log \pi_\theta(\boldsymbol{a}_{t'}, \boldsymbol{s}_{t'}) \nabla_\theta \log \pi_\theta(\boldsymbol{a}_{t'}, \boldsymbol{s}_{t'})^\top \right) r(\boldsymbol{s}_t, \boldsymbol{a}_t) \right] \tag{56}$$

$$= \mathbb{E}_{\boldsymbol{\tau} \sim P_{\mathcal{T}}(\boldsymbol{\tau}|\theta)} \left[ \sum_{t=0}^{H-1} \nabla_\theta \log \pi_\theta(\boldsymbol{a}_t, \boldsymbol{s}_t) \nabla_\theta \log \pi_\theta(\boldsymbol{a}_t, \boldsymbol{s}_t)^\top \left( \sum_{t'=t}^{H-1} r(\boldsymbol{s}_{t'}, \boldsymbol{a}_{t'}) \right) \right] \tag{57}$$

$$= \mathcal{H}_1 \tag{58}$$

By simply exchanging the summation indices $t'$ and $h$ in (55) it is straightforward to show that (55) is the transpose of (54). Hence it is sufficient to show that (54) is equivalent to $\mathcal{H}_{12}$. However, instead of following the direction of the previous proof we will now start with the definition of $\mathcal{H}_{12}$ and derive the expression in (54).

$$\mathcal{H}_{12} = \mathbb{E}_{\boldsymbol{\tau} \sim P_{\mathcal{T}}(\boldsymbol{\tau}|\theta)} \left[ \sum_{t=0}^{H-1} \nabla_\theta \log \pi_\theta(\boldsymbol{a}_t, \boldsymbol{s}_t) \nabla_\theta Q_t^{\pi_\theta}(\boldsymbol{s}_t, \boldsymbol{a}_t)^\top \right] \tag{59}$$

$$\tag{60}$$

The gradient of $Q_t^{\pi_\theta}$ can be expressed recursively:

$$\nabla_\theta Q_t^{\pi_\theta}(\boldsymbol{s}_t, \boldsymbol{a}_t) = \nabla_\theta \mathbb{E}_{\substack{\boldsymbol{s}_{t+1} \\ \boldsymbol{a}_{t+1}}} \left[ Q_{t+1}^{\pi_\theta}(\boldsymbol{s}_{t+1}, \boldsymbol{a}_{t+1}) \right] \tag{61}$$

$$= \mathbb{E}_{\substack{\boldsymbol{s}_{t+1} \\ \boldsymbol{a}_{t+1}}} \left[ \nabla_\theta \log \pi_\theta(\boldsymbol{a}_{t+1}, \boldsymbol{s}_{t+1}) Q_{t+1}^{\pi_\theta}(\boldsymbol{s}_{t+1}, \boldsymbol{a}_{t+1}) + \nabla_\theta Q_{t+1}^{\pi_\theta}(\boldsymbol{s}_{t+1}, \boldsymbol{a}_{t+1}) \right] \tag{62}$$

By induction, it follows that

$$\nabla_\theta Q_t^{\pi_\theta}(\boldsymbol{s}_t, \boldsymbol{a}_t) = \mathbb{E}_{\boldsymbol{\tau}^{t+1:H-1} \sim P_{\mathcal{T}}(\cdot|\theta)} \left[ \sum_{t'=t+1}^{H-1} \nabla_\theta \log \pi_\theta(\boldsymbol{a}_{t'}, \boldsymbol{s}_{t'}) \left( \sum_{h=t'}^{H-1} r(\boldsymbol{s}_h, \boldsymbol{a}_h) \right) \right] \tag{63}$$

When inserting (63) into (59) and swapping the summation, we are able to show that $\mathcal{H}_{12}$ is equivalent to (54).

$$\mathcal{H}_{12} = \mathbb{E}_{\boldsymbol{\tau} \sim P_{\mathcal{T}}(\boldsymbol{\tau}|\theta)} \left[ \sum_{t=0}^{H-1} \sum_{t'=t+1}^{H-1} \nabla_\theta \log \pi_\theta(\boldsymbol{a}_t, \boldsymbol{s}_t) \nabla_\theta \log \pi_\theta(\boldsymbol{a}_{t'}, \boldsymbol{s}_{t'})^\top \left( \sum_{h=t'}^{H-1} r(\boldsymbol{s}_h, \boldsymbol{a}_h) \right) \right] \tag{64}$$

$$= \mathbb{E}_{\boldsymbol{\tau} \sim P_{\mathcal{T}}(\boldsymbol{\tau}|\theta)} \left[ \sum_{t=0}^{H-1} \left( \sum_{t'=0}^{t} \sum_{h=0}^{t'-1} \nabla_\theta \log \pi_\theta(\boldsymbol{a}_{t'}, \boldsymbol{s}_{t'}) \nabla_\theta \log \pi_\theta(\boldsymbol{a}_h, \boldsymbol{s}_h)^\top \right) r(\boldsymbol{s}_t, \boldsymbol{a}_t) \right] \tag{65}$$

This concludes the proof that the hessian of the expected sum of rewards under policy $\pi_\theta$ and an MDP with finite time horizon $H$ can be decomposed into $\mathcal{H}_1 + \mathcal{H}_2 + \mathcal{H}_{12} + \mathcal{H}_{12}^\top$.

$\square$

## B.3 ESTIMATING THE HESSIAN OF THE RL REWARD OBJECTIVE

As pointed out by Al-Shedivat et al. (2018); Stadie et al. (2018) and Foerster et al. (2018), simply differentiating through the gradient of surrogate objective $J^{\text{PGT}}$ as done in the original MAML version (Finn et al., 2017) leads to biased hessian estimates. Specifically, when compared with the unbiased estimate, as derived in (24) and decomposed in Appendix B.2, both $\mathcal{H}_1$ and $\mathcal{H}_{12} + \mathcal{H}_{12}^\top$ are missing. Thus, $\nabla_\theta J_{\text{pre}}$ does not appear in the gradients of the meta-objective (i.e. $\nabla_\theta J = \nabla_\theta J_{\text{post}}$). Only performing gradient descent with $\nabla_\theta J_{\text{post}}$ entirely neglects influences of the pre-update sampling distribution. This issue was overseen in the RL-MAML implementation of Finn et al. (2017). As discussed in Stadie et al. (2018) this leads to poor performance in meta-learning problems that require exploration during the pre-update sampling.

### B.3.1 THE DICE MONTE-CARLO ESTIMATOR

Addressing the issue of incorrect higher-order derivatives of monte-carlo estimators, Foerster et al. (2018) propose DICE which mainly builds upon an newly introduced MagicBox($\square$) operator. This operator allows to formulate monte-carlo estimators with correct higher-order derivatives. A DICE formulation of a policy gradient estimator reads as:

$$J^{\text{DICE}} = \sum_{t=0}^{H-1} \square_\theta(\{a^{t' \leq t}\}) r(s_t, a_t) \tag{66}$$

$$= \sum_{t=0}^{H-1} \exp \left( \sum_{t'=0}^{t} \log \pi_\theta(a_{t'}|s_{t'}) - \bot(\log \pi_\theta(a_{t'}|s_{t'})) \right) r(s_t, a_t) \tag{67}$$

In that, $\bot$ denotes a "stop_gradient" operator (i.e. $\bot(f_\theta(x)) \to f_\theta(x)$ but $\nabla_\theta \bot(f_\theta(x)) \to 0$). Note that $\to$ denotes a "evaluates to" and does not necessarily imply equality w.r.t. to gradients. Hence, $J^{\text{DICE}}(\theta)$ evaluates to the sum of rewards at 0th order but produces the unbiased gradients $\nabla_\theta^n J^{\text{DICE}}(\theta)$ when differentiated n-times (see Foerster et al. (2018) for proof). To shed more light on the maverick DICE formulation, we rewrite (67) as follows:

$$J^{\text{DICE}} = \sum_{t=0}^{H-1} \left( \prod_{t'=0}^{t} \frac{\pi_\theta(a_{t'}|s_{t'})}{\bot(\pi_\theta(a_{t'}|s_{t'}))} \right) r(s_t, a_t) \tag{68}$$

Interpreting this novel formulation, the MagicBox operator $\square_\theta(\{a^{t' \leq t}\})$ can be understood as "dry" importance sampling weight. At 0th order it evaluates to 1 and leaves the objective function unaffected, but when differentiated once it yields an estimator for the marginal rate of return due to a change in the policy-implied trajectory distribution.

In the following we show that on expectation 1) the gradients of (81) match standard policy gradients and 2) its hessian estimate is equal to the hessian of inner RL objective, derived in B.2.

$$\nabla_\theta J^{\text{DICE}} = \sum_{t=0}^{H-1} \nabla_\theta \left( \prod_{t'=0}^{t} \frac{\pi_\theta(a_{t'}|s_{t'})}{\perp(\pi_\theta(a_{t'}|s_{t'}))} \right) r(s_t, a_t) \tag{69}$$

$$= \sum_{t=0}^{H-1} \left( \prod_{t'=0}^{t} \frac{\pi_\theta(a_{t'}|s_{t'})}{\perp(\pi_\theta(a_{t'}|s_{t'}))} \right) \left( \sum_{t'=0}^{t} \nabla_\theta \log \pi_\theta(a_{t'}|s_{t'}) \right) r(s_t, a_t) \tag{70}$$

$$\to \sum_{t=0}^{H-1} \left( \sum_{t'=0}^{t} \nabla_\theta \log \pi_\theta(a_{t'}|s_{t'}) \right) r(s_t, a_t) \tag{71}$$

Here, (71) corresponds to the backward looking credit assignment formulation of policy gradients $\nabla_\theta J^{\text{PGT}}$ as discussed in B.1. Once again we take the derivative in order to obtain the Hessian of $J^{\text{DICE}}$:

$$\nabla_\theta^2 J^{\text{DICE}} = \sum_{t=0}^{H-1} \nabla_\theta \left( \prod_{t'=0}^{t} \frac{\pi_\theta(a_{t'}|s_{t'})}{\perp(\pi_\theta(a_{t'}|s_{t'}))} \right) \left( \sum_{t'=0}^{t} \nabla_\theta \log \pi_\theta(a_{t'}|s_{t'}) \right) r(s_t, a_t) \tag{72}$$

$$+ \left( \prod_{t'=0}^{t} \frac{\pi_\theta(a_{t'}|s_{t'})}{\perp(\pi_\theta(a_{t'}|s_{t'}))} \right) \nabla_\theta \left( \sum_{t'=0}^{t} \nabla_\theta \log \pi_\theta(a_{t'}|s_{t'}) \right) r(s_t, a_t) \tag{73}$$

$$\to \sum_{t=0}^{H-1} \left( \sum_{t'=0}^{t} \nabla_\theta \log \pi_\theta(a_{t'}|s_{t'}) \right) \left( \sum_{t'=0}^{t} \nabla_\theta \log \pi_\theta(a_{t'}|s_{t'}) \right)^\top r(s_t, a_t) \tag{74}$$

$$+ \left( \sum_{t'=0}^{t} \nabla_\theta^2 \log \pi_\theta(a_{t'}|s_{t'}) \right) r(s_t, a_t) \tag{75}$$

In expectation, $\mathbb{E}_{\boldsymbol{\tau} \sim P_{\mathcal{T}}(\boldsymbol{\tau}|\theta)}[\nabla_\theta^2 J^{\text{DICE}}]$ the DICE monte carlo estimate of the hessian is equivalent to the hessian of the inner objective. To show this, we use the expression of $\nabla_\theta^2 J_{\text{inner}}$ (49):

$$\mathbb{E}_{\boldsymbol{\tau} \sim P_{\mathcal{T}}(\boldsymbol{\tau}|\theta)}[\nabla_\theta^2 J^{\text{DICE}}] \tag{76}$$

$$= \mathbb{E}_{\boldsymbol{\tau} \sim P_{\mathcal{T}}(\boldsymbol{\tau}|\theta)} \left[ \sum_{t=0}^{H-1} \left( \sum_{t'=0}^{t} \nabla_\theta \log \pi_\theta(a_{t'}|s_{t'}) \right) \left( \sum_{t'=0}^{t} \nabla_\theta \log \pi_\theta(a_{t'}|s_{t'}) \right)^\top \right. \tag{77}$$

$$\left. r(s_t, a_t) + \left( \sum_{t'=0}^{t} \nabla_\theta^2 \log \pi_\theta(a_{t'}|s_{t'}) \right) r(s_t, a_t) \right] \tag{78}$$

$$= \mathcal{H}_1 + \mathcal{H}_2 + \mathcal{H}_{12} + \mathcal{H}_{12}^\top \tag{79}$$

$$= \nabla_\theta^2 J_{\text{inner}} \tag{80}$$

## B.4 Bias and variance of the curvature estimate

As shown in the previous section, $\nabla_\theta^2 J^{\text{DICE}}$ provides an unbiased estimate of the hessian of the inner objective $J_{\text{inner}} = \mathbb{E}_{\boldsymbol{\tau} \sim P_{\mathcal{T}}(\boldsymbol{\tau}|\theta)}[R(\boldsymbol{\tau})]$. However, recall the DICE objective involves a product of importance weights along the trajectory.

$$J^{\text{DICE}} = \sum_{t=0}^{H-1} \left( \prod_{t'=0}^{t} \frac{\pi_\theta(a_{t'}|s_{t'})}{\perp(\pi_\theta(a_{t'}|s_{t'}))} \right) r(s_t, a_t) \tag{81}$$

Taking the 2nd derivative of this product leads to the outer product of sums in (74) which is of high variance w.r.t to $\boldsymbol{\tau}$. Specifically, this outer product of sums can be decomposed into three terms $\mathcal{H}_1 + \mathcal{H}_{12} + \mathcal{H}_{12}^\top$ (see Appendix B.2). As noted by Furmston et al. (2016), $\mathcal{H}_{12} + \mathcal{H}_{12}^\top$ is particularly difficult to estimate. In section 7.2 we empirically show that the high variance curvature estimates obtained with the DICE objective require large batch sizes and impede sample efficient learning.

In the following we develop a low variance curvature (LVC) estimator $J^{\text{LVC}}$ which matches $J^{\text{DICE}}$ at the gradient level and yields lower-variance estimates of the hessian by neglecting $\mathcal{H}_{12} + \mathcal{H}_{12}^\top$.

Before formally introducing $J^{\text{LVC}}$, we motivate such estimator starting with the policy gradient estimate that was originally derived in Sutton et al. (2000), followed by marginalizing the trajectory level distribution $P_{\mathcal{T}}(\boldsymbol{\tau}|\theta)$ over states $s_t$ and actions $a_t$. Note that we omit reward baselines for notational simplicity.

$$\nabla_\theta J_{\text{inner}} = \mathbb{E}_{\boldsymbol{\tau} \sim P_{\mathcal{T}}(\boldsymbol{\tau}|\theta)} \left[ \sum_{t=0}^{H-1} \nabla_\theta \log \pi_\theta(a_t|s_t) \left( \sum_{t'=t}^{H-1} r(s_{t'}, a_{t'}) \right) \right] \tag{82}$$

$$= \sum_{t=0}^{H-1} \mathbb{E}_{\substack{\boldsymbol{s}_t \sim p_t^{\pi_\theta}(s_t) \\ \boldsymbol{a}_t \sim \pi_\theta(\boldsymbol{a}_t|\boldsymbol{s}_t)}} \left[ \nabla_\theta \log \pi_\theta(a_t|s_t) \left( \sum_{t'=t}^{H-1} r(s_{t'}, a_{t'}) \right) \right] \tag{83}$$

In that, $p_t^{\pi_\theta}(s_t)$ denotes the state visitation frequency at time step $t$, i.e. the probability density of being in $s_t$ after $t$ steps under the policy $\pi_\theta$. In the general case $p_t^{\pi_\theta}(s_t)$ is intractable but depends on the policy parameter $\theta$. We make the simplifying assumption that $p_t^{\pi_\theta}(s_t)$ is fixed in a local region of $\theta$. Since we make this assumption at the gradient level, this corresponds to a 1st order Taylor expansion of $p_t^{\pi_\theta}(s_t)$ in $\theta$. Note that this assumption is also used in the Monotonic Policy Improvement Theory (Kakade & Langford, 2002; Schulman et al., 2015a). Based on this condition, the hessian follows as derivative of (83) whereby a "stop_gradient" expression around the state visitation frequency $p_t^{\pi_\theta}(s_t)$ resembles the 1st order Taylor approximation:

$$\mathbb{E}_{\boldsymbol{\tau}} \left[ \nabla_\theta^2 J^{\text{LVC}} \right] = \nabla_\theta \sum_{t=0}^{H-1} \mathbb{E}_{\substack{\boldsymbol{s}_t \sim \perp(p_t^{\pi_\theta}(s_t)) \\ \boldsymbol{a}_t \sim \pi_\theta(\boldsymbol{a}_t|\boldsymbol{s}_t)}} \left[ \nabla_\theta \log \pi_\theta(a_t|s_t) \left( \sum_{t'=t}^{H-1} r(s_{t'}, a_{t'}) \right) \right] \tag{84}$$

$$= \sum_{t=0}^{H-1} \mathbb{E}_{\substack{\boldsymbol{s}_t \sim \perp(p_t^{\pi_\theta}(s_t)) \\ \boldsymbol{a}_t \sim \pi_\theta(\boldsymbol{a}_t|\boldsymbol{s}_t)}} \left[ \nabla_\theta \log \pi_\theta(a_t|s_t) \nabla_\theta \log \pi_\theta(a_t|s_t)^\top \left( \sum_{t'=t}^{H-1} r(s_{t'}, a_{t'}) \right) \right. \tag{85}$$

$$\left. + \nabla_\theta^2 \log \pi_\theta(a_t|s_t) \left( \sum_{t'=t}^{H-1} r(s_{t'}, a_{t'}) \right) \right] \tag{86}$$

Since the expectation in (84) is intractable it must be evaluated by a monte carlo estimate. However, simply replacing the expectation with an average of samples trajectories induces a wrong hessian that does not correspond to (86) since outer product of log-gradients would be missing when differentiated. To ensure that automatic differentiation still yields the correct hessian, we add a "dry" importance weight comparable to DICE:

$$\nabla_\theta J^{\text{LVC}} = \sum_{t=0}^{H-1} \frac{\pi_\theta(a_t|s_t)}{\perp(\pi_\theta(a_t|s_t))} \nabla_\theta \log \pi_\theta(a_t|s_t) \left( \sum_{t'=t}^{H-1} r(s_{t'}, a_{t'}) \right) \quad \boldsymbol{\tau} \sim P_{\mathcal{T}}(\boldsymbol{\tau}|\theta) \tag{87}$$

When integrated this resembles the LVC "surrogate" objective $J^{\text{LVC}}$.

$$J^{\text{LVC}} = \sum_{t=0}^{H-1} \frac{\pi_\theta(a_t|s_t)}{\perp(\pi_\theta(a_t|s_t))} \left( \sum_{t'=t}^{H-1} r(s_{t'}, a_{t'}) \right) \quad \boldsymbol{\tau} \sim P_{\mathcal{T}}(\boldsymbol{\tau}|\theta) \tag{88}$$

The gradients of $J^{\text{LVC}}$ match $\nabla_\theta J^{\text{DICE}}$ and resemble an unbiased policy gradient estimate:

$$\nabla_\theta J^{\text{LVC}} = \sum_{t=0}^{H-1} \frac{\nabla_\theta \pi_\theta(a_t|s_t)}{\perp(\pi_\theta(a_t|s_t))} \left( \sum_{t'=t}^{H-1} r(s_{t'}, a_{t'}) \right) \tag{89}$$

$$= \sum_{t=0}^{H-1} \frac{\pi_\theta(a_t|s_t)}{\perp(\pi_\theta(a_t|s_t))} \nabla_\theta \log \pi_\theta(a_t|s_t) \left( \sum_{t'=t}^{H-1} r(s_{t'}, a_{t'}) \right) \tag{90}$$

$$\rightarrow \sum_{t=0}^{H-1} \nabla_\theta \log \pi_\theta(a_t|s_t) \left( \sum_{t'=t}^{H-1} r(s_{t'}, a_{t'}) \right) \tag{91}$$

The respective Hessian can be obtained by differentiating (90):

$$\nabla_\theta^2 J^{\text{LVC}} = \nabla_\theta \sum_{t=0}^{H-1} \frac{\pi_\theta(a_t|s_t)}{\perp(\pi_\theta(a_t|s_t))} \nabla_\theta \log \pi_\theta(a_t|s_t) \left( \sum_{t'=t}^{H-1} r(s_{t'}, a_{t'}) \right) \tag{92}$$

$$= \sum_{t=0}^{H-1} \frac{\pi_\theta(a_t|s_t)}{\perp(\pi_\theta(a_t|s_t))} \nabla_\theta \log \pi_\theta(a_t|s_t) \nabla_\theta \log \pi_\theta(a_t|s_t)^\top \left( \sum_{t'=t}^{H-1} r(s_{t'}, a_{t'}) \right) \tag{93}$$

$$+ \frac{\pi_\theta(a_t|s_t)}{\perp(\pi_\theta(a_t|s_t))} \nabla_\theta^2 \log \pi_\theta(a_t|s_t) \left( \sum_{t'=t}^{H-1} r(s_{t'}, a_{t'}) \right) \tag{94}$$

$$\rightarrow \sum_{t=0}^{H-1} \nabla_\theta \log \pi_\theta(a_t|s_t) \nabla_\theta \log \pi_\theta(a_t|s_t)^\top \left( \sum_{t'=t}^{H-1} r(s_{t'}, a_{t'}) \right) \tag{95}$$

$$+ \nabla_\theta^2 \log \pi_\theta(a_t|s_t) \left( \sum_{t'=t}^{H-1} r(s_{t'}, a_{t'}) \right) \tag{96}$$

$$= \sum_{t=0}^{H-1} \left( \sum_{t'=0}^{t} \nabla_\theta \log \pi_\theta(a_{t'}|s_{t'}) \nabla_\theta \log \pi_\theta(a_t|s_t)^\top \right) r(s_t, a_t) \tag{97}$$

$$+ \left( \sum_{t'=0}^{t} \nabla_\theta^2 \log \pi_\theta(a_{t'}|s_{t'}) \right) r(s_t, a_t) \tag{98}$$

In expectation $\nabla_\theta^2 J^{\text{LVC}}$ is equivalent to $\mathcal{H}_1 + \mathcal{H}_2$:

$$\mathbb{E}_{\boldsymbol{\tau} \sim P_\mathcal{T}(\boldsymbol{\tau}|\theta)} \left[ J^{\text{LVC}} \right] = \mathbb{E}_{\boldsymbol{\tau} \sim P_\mathcal{T}(\boldsymbol{\tau}|\theta)} \left[ \sum_{t=0}^{H-1} \left( \sum_{t'=0}^{t} \nabla_\theta \log \pi_\theta(\boldsymbol{a}_{t'}|\boldsymbol{s}_{t'}) \nabla_\theta \log \pi_\theta(\boldsymbol{a}_t|\boldsymbol{s}_t)^\top \right) r(\boldsymbol{s}_t, \boldsymbol{a}_t) \right] \tag{99}$$

$$+ \mathbb{E}_{\boldsymbol{\tau} \sim P_\mathcal{T}(\boldsymbol{\tau}|\theta)} \left[ \sum_{t=0}^{H-1} \left( \sum_{t'=0}^{t} \nabla_\theta^2 \log \pi_\theta(\boldsymbol{a}_{t'}|\boldsymbol{s}_{t'}) \right) r(\boldsymbol{s}_t, \boldsymbol{a}_t) \right] \tag{100}$$

$$= \mathcal{H}_1 + \mathcal{H}_2 \tag{101}$$

The Hessian $\nabla_\theta^2 J^{\text{LVC}}$ no longer provides an unbiased estimate of $\nabla_\theta^2 J_{\text{inner}}$ since neglects the matrix term $\mathcal{H}_{12} + \mathcal{H}_{12}^\top$. This approximation is based on the assumption that the state visitation distribution is locally unaffected by marginal changes in $\theta$ and leads to a substantial reduction of variance in the hessian estimate. Furmston et al. (2016) show that under certain conditions (i.e. infinite horizon MDP, sufficiently rich policy parameterisation) the term $\mathcal{H}_{12} + \mathcal{H}_{12}^\top$ vanishes around a local optimum $\theta^*$. Given that the conditions hold, this implies that $\mathbb{E}_\tau[\nabla_\theta^2 J^{\text{LVC}}] \rightarrow \mathbb{E}_\tau[\nabla_\theta^2 J^{\text{DICE}}]$ as $\theta \rightarrow \theta^*$, i.e. the bias of the LCV estimator becomes negligible close to the local optimum. The experiments in section 7.2 confirm this theoretical argument empirically and show that using the low variance curvature estimates obtained through $J^{\text{LVC}}$ improve the sample-efficiency of meta-learning by a significant margin.

## C PROXIMAL POLICY SEARCH METHODS

### C.1 MONOTONIC POLICY IMPROVEMENT THEORY

This section provides a brief introduction to policy performance bounds and the theory of monotonic policy improvement in the setting of reinforcement learning. While Section 6 discusses the extension of this theory to meta learning, the following explanations assume a standard RL setting where $\mathcal{T}$ is exogenously given. Hence, we will omit mentioning the dependence on $\mathcal{T}$ for notational brevity. Since the monotonic policy improvement frameworks relies on infinite-time horizon MDPs, we assume $H \rightarrow \infty$ for the remainder of this chapter.

In addition to the expected reward $J(\pi)$ under policy $\pi$, we will use the state value function $V^\pi$, the state-action value function $Q^\pi$ as well as the advantage function $A^\pi$:

$$V^\pi(s) = \mathbb{E}_{\boldsymbol{a}_0, \boldsymbol{s}_1, \dots} \left[ \sum_{t=0}^\infty \gamma^t r(\boldsymbol{s}_t, \boldsymbol{a}_t) \middle| \boldsymbol{s}_t = s \right]$$

$$Q^\pi(s, a) = \mathbb{E}_{\boldsymbol{s}_1, \boldsymbol{a}_1, \dots} \left[ \sum_{t=0}^\infty \gamma^t r(\boldsymbol{s}_t, \boldsymbol{a}_t) \middle| \boldsymbol{s}_t = s, \boldsymbol{a}_0 = a \right] = r(s, a) + \gamma \mathbb{E}_{s' \sim p(s'|s, a)} \left[ V_\pi(s') \right]$$

$$A^\pi(s, a) = Q^\pi(s, a) - V^\pi(s)$$

with $\boldsymbol{a}_t \sim \pi(a_t|s_t)$ and $\boldsymbol{s}_{t+1} \sim p(s_{t+1}|s_t, a_t)$.

The expected return under a policy $\tilde{\pi}$ can be expressed as the sum of the expected return of another policy $\pi$ and the expected discounted advantage of $\tilde{\pi}$ over $\pi$ (see Schulman et al. (2015a) for proof).

$$J(\tilde{\pi}) = J(\pi) + \mathbb{E}_{\boldsymbol{\tau} \sim P(\boldsymbol{\tau}, \tilde{\pi})} \left[ \sum_{t=0}^\infty \gamma^t A^\pi(\boldsymbol{s}_t, \boldsymbol{a}_t) \right]$$

Let $d_\pi$ denote the discounted state visitation frequency:

$$d_\pi(\boldsymbol{s}) = \gamma_t \sum_{t=0}^\infty p(s_t = \boldsymbol{s}|\pi)$$

We can use $d_\pi$ to express the expectation over trajectories $\boldsymbol{\tau} \sim p^\pi(\tau)$ in terms of states and actions:

$$J(\tilde{\pi}) = J(\pi) + \mathbb{E}_{\substack{\boldsymbol{s} \sim d_{\tilde{\pi}}(\boldsymbol{s}) \\ \boldsymbol{a} \sim \tilde{\pi}(\boldsymbol{a}|\boldsymbol{s})}} \left[ A^\pi(\boldsymbol{s}, \boldsymbol{a}) \right] \tag{102}$$

Local policy search aims to find a policy update $\pi \to \tilde{\pi}$ in the proximity of $\pi$ so that $J(\tilde{\pi})$ is maximized. Since $J(\pi)$ is not affected by the policy update $\pi \to \tilde{\pi}$, it is sufficient to maximize the expected advantage under $\tilde{\pi}$. However, the complex dependence of $d_{\tilde{\pi}}(\boldsymbol{s})$ on $\tilde{\pi}$ makes it hard to directly maximize the objective in (102). Using a local approximation of (102) where it is assumed that the state visitation frequencies $d_\pi$ and $d_{\tilde{\pi}}$ are identical, the optimization can be phrased as

$$\tilde{J}_\pi(\tilde{\pi}) = J(\pi) + \mathbb{E}_{\substack{\boldsymbol{s} \sim d_\pi(\boldsymbol{s}) \\ \boldsymbol{a} \sim \tilde{\pi}(\boldsymbol{a}|\boldsymbol{s})}} \left[ A^\pi(\boldsymbol{s}, \boldsymbol{a}) \right] = J(\pi) + \mathbb{E}_{\substack{\boldsymbol{s} \sim d_\pi(\boldsymbol{s}) \\ \boldsymbol{a} \sim \pi(\boldsymbol{a}|\boldsymbol{s})}} \left[ \frac{\tilde{\pi}(\boldsymbol{a}|\boldsymbol{s})}{\pi(\boldsymbol{a}|\boldsymbol{s})} A^\pi(\boldsymbol{s}, \boldsymbol{a}) \right] \tag{103}$$

In the following we refer to $\tilde{J}(\tilde{\pi})$ as surrogate objective. It can be shown that the surrogate objective $\tilde{J}$ matches $J$ to first order when $\pi = \tilde{\pi}$ (see Kakade & Langford (2002)). If $\pi_\theta$ is a parametric and differentiable function with parameter vector $\theta$, this means that for any $\theta_o$:

$$\tilde{J}_{\pi_{\theta_o}}(\pi_{\theta_o}) = J_{\pi_{\theta_o}}(\pi_{\theta_o}) \quad \text{and} \quad \nabla_\theta \tilde{J}_{\pi_{\theta_o}}(\pi_\theta)\big|_{\theta_o} = \nabla_\theta J_{\pi_{\theta_o}}(\pi_\theta)\big|_{\theta_o} \tag{104}$$

When $\pi \neq \tilde{\pi}$, an approximation error of the surrogate objective $\tilde{J}$ w.r.t. to the true objective $J$ is introduced. Achiam et al. (2017) derive a lower bound for the true expected return of $\tilde{\pi}$:

$$J(\tilde{\pi}) \geq J_\pi(\tilde{\pi}) - C\sqrt{\mathbb{E}_{\boldsymbol{s} \sim d_\pi} \left[ \mathcal{D}_{\mathrm{KL}}[\tilde{\pi}(\cdot|\boldsymbol{s})||\pi(\cdot|\boldsymbol{s})] \right]} = J_\pi(\tilde{\pi}) - C\sqrt{\bar{\mathcal{D}}_{\mathrm{KL}}[\tilde{\pi}||\pi]} \tag{105}$$

with $C = \frac{\sqrt{2}\gamma}{1-\gamma} \max_s |\mathbb{E}_{\boldsymbol{a} \sim \tilde{\pi}(\boldsymbol{a}, s)}[A^\pi(s, \boldsymbol{a})]|$

## C.2 TRUST REGION POLICY OPTIMIZATION (TRPO)

Trust region policy optimization (TPRO) (Schulman et al., 2015a) attempts to approximate the bound in (105) by phrasing local policy search as a constrained optimization problem:

$$\arg\max_\theta \mathbb{E}_{\substack{\boldsymbol{s} \sim d_{\pi_{\theta_o}}(\boldsymbol{s}) \\ \boldsymbol{a} \sim \pi_{\theta_o}(\boldsymbol{a}|\boldsymbol{s})}} \left[ \frac{\pi_\theta(\boldsymbol{a}|\boldsymbol{s})}{\pi_{\theta_o}(\boldsymbol{a}|\boldsymbol{s})} A^{\pi_{\theta_o}}(\boldsymbol{s}, \boldsymbol{a}) \right] \quad \text{s.t.} \quad \bar{\mathcal{D}}_{\mathrm{KL}}[\pi_{\theta_o}||\pi_\theta] \leq \delta \tag{106}$$

Thereby the KL-constraint $\delta$ induces a local trust region around the current policy $\pi_{\theta_o}$. A practical implementation of TPRO uses a quadratic approximation of the KL-constraint which leads to the following update rule:

$$\theta \leftarrow \theta + \sqrt{\frac{2\delta}{g^\top F g}} F^{-1} g \tag{107}$$

with $g := \nabla_\theta \mathbb{E}_{\substack{s \sim d_{\pi_{\theta_o}}(s) \\ a \sim \pi_{\theta_o}(a|s)}} \left[ \frac{\pi_\theta(a|s)}{\pi_{\theta_o}(a|s)} A^{\pi_{\theta_o}}(s, a) \right]$ being the gradient of the objective and $F = \nabla_\theta^2 \bar{\mathcal{D}}_{\mathrm{KL}}[\pi_{\theta_o} || \pi_\theta]$ the Fisher information matrix of the current policy $\pi_{\theta_o}$. In order to avoid the cubic time complexity that arise when inverting $F$, the Conjugate Gradient (CG) algorithm is typically used to approximate the Hessian vector product $F^{-1}g$.

### C.3 Proximal Policy Optimization (PPO)

While TPRO is framed as constrained optimization, the theory discussed in Appendix C.1 suggest to optimize the lower bound. Based on this insight, Schulman et al. (2017) propose adding a KL penalty to the objective and solve the following unconstrained optimization problem:

$$\arg\max_\theta \; \mathbb{E}_{\substack{s \sim d_{\pi_{\theta_o}}(s) \\ a \sim \pi_{\theta_o}(a|s)}} \left[ \frac{\pi_\theta(a|s)}{\pi_{\theta_o}(a|s)} A^{\pi_{\theta_o}}(s, a) - \beta \mathcal{D}_{\mathrm{KL}}[\pi_{\theta_o}(\cdot|s) || \pi_\theta(\cdot|s)] \right] \tag{108}$$

However, they also show that it is not sufficient to set a fixed penalty coefficient $\beta$ and propose two alternative methods, known as Proximal Policy Optimization (PPO) that aim towards alleviating this issue:

1) Adapting the KL coefficient $\beta$ so that a desired target KL-divergence $\bar{\mathcal{D}}_{\mathrm{KL}}[\pi_{\theta_o} || \pi_\theta]$ between the policy before and after the parameter update is achieved

2) Clipping the likelihood ratio so that the optimization has no incentive to move the policy $\pi_\theta$ too far away from the original policy $\pi_{\theta_o}$. A corresponding optimization objective reads as:

$$J^{\mathrm{CLIP}} = \mathbb{E}_{\substack{s \sim d_{\pi_{\theta_o}}(s) \\ a \sim \pi_{\theta_o}(a|s)}} \left[ \min \left( \frac{\pi_\theta(a|s)}{\pi_{\theta_o}(a|s)} A^{\pi_{\theta_o}}(s, a) , \; \mathrm{clip}_{1-\epsilon}^{1+\epsilon} \left( \frac{\pi_\theta(a|s)}{\pi_{\theta_o}(a|s)} \right) A^{\pi_{\theta_o}}(s, a) \right) \right] \tag{109}$$

Empirical results show that the latter approach leads to better learning performance (Schulman et al., 2017).

Since PPO objective keeps $\pi_\theta$ in proximity of $\pi_{\theta_o}$, it allows to perform multiple gradient steps without re-sampling trajectories from the updated policy. This property substantially improves the data-efficiency of PPO over vanilla policy gradient methods which need to re-estimate the gradients after each step.

## D    Experiments

### D.1    Hyperparameter Choice

The optimal hyperparameter for each algorithm was determined using parameter sweeps. Table 1 contains the hyperparameter settings used for the different algorithms. Any environment specific modifications are noted in the respective paragraph describing the environment.

### D.2    Environment Specifications

**PointEnv** (used in the experiment in 7.3)

- Trajectory Length : 100
- Num Adapt Steps : 3

| All Algorithms | |
|---|---|
| Policy Hidden Layer Sizes | (64, 64) |
| | (128, 128) for Humanoid |
| Num Adapt Steps | 1 |
| Inner Step Size $\alpha$ | 0.01 |
| Tasks Per Iteration | 40 |
| Trajectories Per Task | 20 |
| **ProMP** | |
| Outer Learning Rate $\beta$ | 0.001 |
| Grad Steps Per ProMP Iteration | 5 |
| Outer Clip Ratio $\epsilon$ | 0.3 |
| KL Penalty Coef. $\eta$ | 0.0005 |
| **MAML-TRPO** | |
| Trust Region Size | 0.01 |
| **MAML-VPG** | |
| Outer Learning Rate $\beta$ | 0.001 |

Table 1: Hyperparameter settings used in each algorithm

In this environment, each task corresponds to one corner of the area. The point mass must reach the goal by applying directional forces. The agent only experiences a reward when within a certain radius of the goal, and the magnitude of the reward is equal to the distance to the goal.

**HalfCheetahFwdBack, AntFwdBack, WalkerFwdBack, HumanoidFwdBack**

- Trajectory Length : 100 (HalfCheetah, Ant); 200 (Humanoid, Walker)
- Num Adapt Steps: 1

The task is chosen between two directions - forward and backward. Each agent must run along the goal direction as far as possible, with reward equal to average velocity minus control costs.

**AntRandDirec, HumanoidRandDirec**

- Trajectory Length : 100 (Ant); 200 (Humanoid)
- Num Adapt Steps: 1

Each task corresponds to a random direction in the XY plane. As above, each agent must learn to run in that direction as far as possible, with reward equal to average velocity minus control costs.

**AntRandGoal**

- Trajectory Length : 200
- Num Adapt Steps: 2

In this environment, each task is a location randomly chosen from a circle in the XY plane. The goal is not given to the agent - it must learn to locate, approach, and stop at the target. The agent receives a penalty equal to the distance from the goal.

**HopperRandParams, WalkerRandParams**

- Trajectory Length : 200
- Inner LR : 0.05
- Num Adapt Steps: 1

The agent must move forward as quickly as it can. Each task is a different randomization of the simulation parameters, including friction, joint mass, and inertia. The agent receives a reward equal to its velocity.

### D.3 FURTHER EXPERIMENTS RESULTS

In addition to the six environments displayed in 2, we ran experiments on the other four continuous control environments described above. The results are displayed in 7.

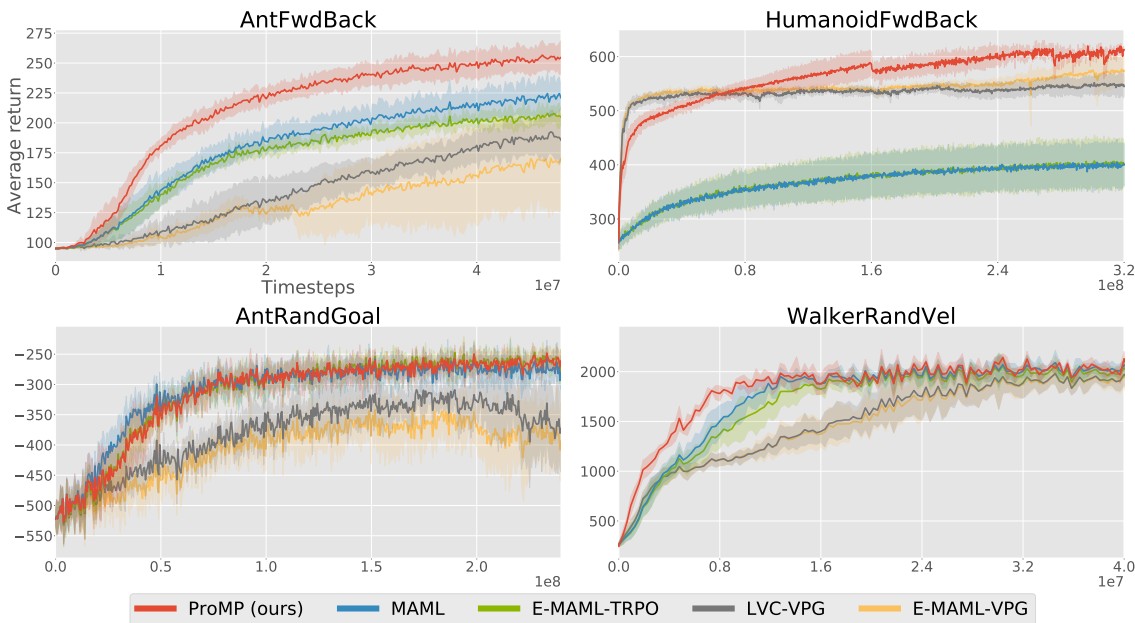

Figure 7: Meta-learning curves of ProMP and four other gradient-based meta-learning algorithms in four new Mujoco environments

In addition to the improved sample complexity and better asymptotic performance, another advantage of ProMP is its computation time. Figure 8 shows the average time spent per iteration throughout the learning process in the humanoid environment differences of ProMP, LVC-VPG, and MAML-TRPO. Due to the expensive conjugate gradient steps used in TRPO, MAML takes far longer than either first order method. Since ProMP takes multiple stochastic gradient descent steps per iteration, it leads to longer outer update times compared to VPG, but in both cases the update time is a fraction of the time spent sampling from the environment.

The difference in sampling time is due to the reset process: resetting the environment when the agent "dies" is an expensive operation. ProMP acquires better performance quicker, and as a result the agent experiences longer trajectories and the environment is reset less often. In our setup, instances of the environment are run in parallel and performing a reset blocks all environments.

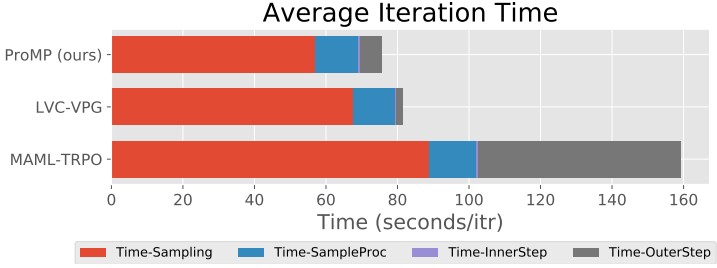

Figure 8: Comparison of wall clock time with different algorithms on HumanoidRandDirec, averaged over all iterations

