# OpenReview forum: "ProMP: Proximal Meta-Policy Search"
_ICLR.cc/2019/Conference_

### Official Review · AnonReviewer1 · 2018-11-02
**Strong paper, Strong accept**

**Rating:** 9
**Confidence:** 3

**Review:**

The paper first examines the objective function optimized in MAML and E-MAML and interprets the terms as different credit assignment criteria. MAML takes into account the dependences between pre-update trajectory and pre-update policy, post-update trajectory and post-update policy by forcing the gradient of the two policies to be aligned, which results in better learning properties.
Thought better, the paper points out MAML has incorrect estimation for the hessian in the objective. To address that, the paper propose a low variance curvature estimator (LVC). However, naively solving the new objective with LVC with TRPO is computationally prohibitive. The paper addresses this problem by proposing an objective function that combines PPO and a slightly modified version of LVC.

Quality: strong, clarity:strong, originality:strong, significance: strong,

Pros:
- The paper provides strong theoretical results. Though mathematically intense, the paper is written quite well and is easy to follow.
- The proposed method is able to improve in sample complexity, speed and convergence over past methods.
- The paper provides strong empirical results over MAML, E-MAML. They also show the effective of the LVC objective by comparing LVC over E-MAML using vanilla gradient update.
- Figure 4 is particularly interesting. The results show different exploration patterns used by different method and is quite aligned with the theory.
Cons:
- It would be nice to add more comparison and analysis on the variance. Since LVC is claimed to reduce variance of the gradient, it would be nice to show more empirical evidences that supports this. (By looking at Figure 2, although not directly related, LVC-VPG seems to have pretty noisy behaviour)

---

> ### Author Response · Authors · 2018-11-25
> **Thank you for your feedback!**
>
> We thank the reviewer for the valuable feedback. Indeed, the LVC-VPG learning curves exhibit high noise. We believe that the bias introduced by LVC makes the learning less stable when VPG is used as outer optimizer. However,  the mechanisms in ProMP that ensure proximity w.r.t. to the policy’s KL-divergence may counteract these instabilities, explaining why ProMP works so well in practice.
>
> Following the suggestion of the reviewer we have extended our comparison and the analysis of the variance. In particular, we added learning curves for DiCE-VPG to the benchmarks in section 7.1. Furthermore, we have extended the analysis of the variance to more environments and and more training iterations. The result show that LVC has a substantially higher data-efficiency and its meta-gradients consistently exhibit a lower variance than DiCE.
>
> We hope that this results further underpin the soundness of our claims and show the importance of our method.

---

### Official Review · AnonReviewer3 · 2018-11-05

**Rating:** 7
**Confidence:** 3

**Review:**

In this paper, the author proposed an efficient surrogate loss for estimating  Hessian in the setting of Meta-reinforcement learning (Finn.et al, 2017), which significantly reduce the variance while introducing small bias. The author verified their proposed method with other meta-learning algorithms on the Mujoco benchmarks. The author also compared with unbiased higher order gradient estimation method-DiCE in terms of gradient variance and average return.

The work is essentially important due to the need for second-order gradient estimation for meta-learning (Finn et al., 2017) and other related work such as multi-agent RL. The results look promising and the method is easy to implement. I have two detail questions about the experiment:

1) As the author states, the new proposed method introduces bias while reducing variance significantly. It is necessary to examine the MSE, Bias, Variance of the gradient estimatorsquantitatively  for the proposed and related baseline methods (including MAML, E-MAML-TRPO, LVC-VPG, etc). If the bias is not a big issue empirically, the proposed method is good to use in practice.

2)  The author should add DiCE in the benchmark in section 7.1, which will verify its advantage over DiCE thoroughly.

Overall this is a good paper and I vote for acceptance.


Finn, Chelsea, et al. "Model-agnostic meta-learning for fast adaptation of deep networks." ICML 2017.

Foerster, Jakob, et al. "DiCE: The Infinitely Differentiable Monte-Carlo Estimator." ICML 2018.

---

> ### Author Response · Authors · 2018-11-25
> **Thank you for your feedback!**
>
> We thank the reviewer for the valuable feedback provided. As suggested by the reviewer, we have added more experiments in order to underpin the advantage of the LVC over the DiCE estimator.
>
> In particular, we have included DiCE-VPG in the benchmark section 7.1. Results in all environments demonstrate that the learning performance of DiCE is inferior to LVC. In many of the environments, DiCE learns very slowly when compared to the other methods. We ascribe the poor learning performance of DiCE to the high variance of its meta-gradient estimates.
> To further strengthen this hypothesis, we have extended the meta-gradient variance experiments to more environments and more training iterations.
>
> All in all, the bias introduced by LVC seems to make the learning a little bit more unstable when VPG is used as outer optimizer. However, the gains in data-efficiency substantially outwage this disadvantage. Ultimately, the mechanisms in ProMP that ensure proximity w.r.t. to the policy’s KL-divergence may counteract these instabilities during training, giving us a stable and efficient meta-learning algorithm.
>
> We hope that the experiments and discussions, added to the paper, further substantiate the soundness of our claims.

---

### Official Review · AnonReviewer2 · 2018-11-08
**an interesting trial to correct the current algorithm, but weak support to the claim**

**Rating:** 6
**Confidence:** 3

**Review:**


In this paper, the authors investigate the gradient calculation in the original MAML (Finn et al. 2017) and E-MAML (Al-Shedivat et al. 2018). By comparing the differences in the gradients of these two algorithms, the authors demonstrate the advantages of the original MAML in taking the casual dependence into account. To obtain the correct estimation of the gradient through auto-differentiation, the authors exploit the DiCE formulation. Considering the variance in the DiCE objective formulation, the authors finally propose an objective which leads to low-variance but biased gradient. The authors verify the proposed methods in meta-RL tasks and achieves comparable performances to MAML and E-MAML.


Although the ultimate algorithm proposed by this paper is not far away from MAML and E-MAML, they did a quite good job in clarify the differences in the existing variants of MAML from the gradient computation perspective and reveal the potential error due to the auto-differentiation. The proposed new objective and the surrogate is well-motivated from such observation and the trade-off between variance and bias.


My major concern is how big the effect is if we use (3) comparing to (4) in calculate the gradient. As the authors showed, the only difference between (3) and (4) is the weights in front of the term \nabla_\theta\log\pi_\theta: the E-MAML is a fixed weight and the MAML is using a adaptive through the inner product. Whether the final difference in Figure 4 between MAML and E-MAML is all caused by such difference in gradient estimation is not clearly. In fact, based on the other large-scale high-dimension empirical experiments in Figure 2, it seems the difference in gradient estimator (3) and (4) does not induced too much difference in final performances between MAML and E-MAML. Based on such observation, I was wondering the consistent better performance of the proposed algorithm might not because the corrected gradient computation from the proposed objective. It might because the clip operation or other components in the algorithm. To make a more convincing argument, it will be better if the authors can evaluate different gradient within the same updates.

I am willing to raise my score if the author can address the question.

minor:

The gradients calculation in Eq (2) and (3) are not consistent with the Algorithm and the appendix.

The notation is not consistent with common usage: \nabla^2 is actually used for denoting the Laplace operator, i.e., \nabla^2 = \nabla \cdot \nable, which is a scalar.

---

> ### Author Response · Authors · 2018-11-25
> **Thank you for your feedback!**
>
> We thank the reviewer for the valuable feedback. The main concern of the reviewer is that the difference in performance between using equation (4) and (3) is not as significant as we claim.
>
> First, we want to clarify what might be a misunderstanding. The results labeled as MAML in Fig. 2 are obtained using the original MAML implementation, which, due to the use of a normal score function estimator, computes the wrong meta-gradient instead of the one given by Eq. 3 (you can find a discussion on this in section 5, further elaborated in the appendix). We have added some further explanations in section 5 to further clarify this.
> Overall, here is a legend for what each name refers to: :
> MAML: no pre-adaptation credit assignment, i.e.  \nabla J = \nabla J_post, i.e. how MAML was implemented for the original MAML paper (but this actually doesn’t follow the math correctly)
> E-MAML: naive pre-adaptation credit assignment as in Eq. 3
> DiCE: (unbiased but high variance) credit assignment as in Eq. 4
> LVC: (slightly biased but low variance) credit assignment as in Eq. 4
> ProMP: our final method (described in section 6)
>
> With the nomenclature clarified, let us highlight how our experiments showcase the difference w.r.t the credit assignment.
>
> First, we have added a plot showing the effect of each formulation when the same optimizer is used (see Figure 3). We performed this experiment as an ablation study in order eliminate possible influences of the outer optimizer, i.e. PPO and TRPO. These results consistently show the superior performance of the low variance version of Eq. 4 (LVC-VPG) when compared with Eq. 3 (E-MAML-VPG). Due to the high variance nature of Eq. 4 (DiCE-VPG) its performance saturates below the other formulations. This effect is discussed in section 7.2.
>
> Second, the experiment in Fig. 5 illustrates the differences w.r.t. the meta-learned pre-adaptation policy behavior. Since MAML does not assign any credit to the pre-adaptation policy, it fails so solve the task. Though, E-MAML (Eq. 3) is able to solve the task, it does not learn an effective task identification policy since it can only assign credit to batches of pre-adaptation trajectories. In contrast, LVC (Eq. 4) can assign credit to individual pre-adaptation trajectories which is reflected by its superior task identification behavior.
>
> Third, the fact that the results of MAML and E-MAML in Fig. 2 and Fig. 3 are comparable underpins the ineffectiveness of the naive credit assignment: there is little difference between zero pre-adaptation credit assignment and the credit assignment of E-MAML (discussed in section 4).
>
> Finally, the experiment in Fig. 6 depicts computed gradients and convergence properties corresponding to Eq. 3 and Eq. 4 in a simple toy environment. Once more, this experiment shows the advantage of formulation I over formulation II.
>
> We have clarified this in the experiment sections of the paper, and will be happy to add further
> clarifications if the reviewer requests it.

---

### Public Comment · (anonymous) · 2018-10-10
**typo in equation 43**

Hey, is there a typo in equation 43? I mean on the left side there should be a Hessian instead of an expectation of a Hessian.

---

> ### Author Response · Authors · 2018-10-15
> **typo in equation 43**
>
> Thanks for pointing out the typo. Indeed, there should be no expectation around J_inner, since J_inner is already an expectation itself. This has no mathematical implications but is unnecessary. We have fixed the denoted typo as well as further minor typos we have found in the appendix. The changes will appear in the pdf as soon as we are able to update it during the rebuttal.

---

> > ### Public Comment · (anonymous) · 2018-10-23
> > **thanks and question about hessian**
> >
> > Hey, thanks for your response.
> >
> > In your submission, you argue that DICE need calculation of the outer product of sum, which leads to high variance, then you guys propose a method to somehow get rid of it, what I don't understand is why not use something like `'tf.hessians' to analytical compute the gradient.

---

> > > ### Author Response · Authors · 2018-10-23
> > > **why not just implementing the meta-gradients directly**
> > >
> > > Indeed, it would be possible to code up the analytical expression of the LVC gradients (i.e. the terms H_1 and H_2 as derived in Equation 99 and 100). However, to do so is tedious and error prone. Researchers / engineers usually prefer to implement a “surrogate loss” such as the LVC objective which is more elegant and clean. Furthermore, it is not straightforward to code up the LVC gradients with tensorflow since tf.gradients and tf.hessians does not return a batch of gradients / hessians but instead the sum of gradients over the batch (see https://www.tensorflow.org/api_docs/python/tf/gradients). Thus we are not aware of any efficient way of computing H_2 (outer product of grad log_probs) in batch.

---

> > > > ### Public Comment · (anonymous) · 2018-10-26
> > > > **thanks for reply and question about experiment settings**
> > > >
> > > > Thanks for your responding!
> > > >
> > > > A question about your experiment settings, namely, what does the 'MAML' in Figure 2. means? MAML+TRPO or MAML+TRPO+DICE?

---

### Public Comment · (anonymous) · 2018-11-07
**Questions about experiments**

Hello, I have some questions about experiments in the paper.

I'd like to clarify my understanding of the experimental setup for results in Fig. 2.
Are the returns plotted on the y-axis averaged over the the tasks that were sampled *for meta-training* during the most recent update? If so, then how/where is the performance of meta-testing measured?

For FwdBack environments:
If appears that the task distribution is rather simple, containing only two tasks. Is this correct?
If yes, then during meta-training the method quickly sees all possible tasks from the training distribution and so no task at any meta-test time would be novel at all. How can one interpret the utility of these environments for evaluating Meta-RL methods?

Thanks.

---

> ### Author Response · Authors · 2018-11-28
> **Re: Questions about experiments**
>
> Thanks a lot for the excellent comment!
>
> The returns in Fig. 2 are estimated by sampling a batch of tasks from the task distribution and rolling out a number of trajectories with the adapted policy in each of the tasks. Then we average over all sampled tasks, trajectories and seeds.
>
> Regarding meta-testing, gradient-based meta-learning methods just provide guarantees of adaptation after one gradient step (or a few if you meta-trained for it). This is exactly what we measure our our benchmarks on throughout the meta-training process. Nevertheless, we agree that it is important and interesting to evaluate how the method performs after more than one adaptation step is performed, and how it behaves in out-of-distribution tasks. These results will be added in the camera ready, and we can clarify any questions regarding these experiments.
>
> The FwdBackw environments have been used as benchmark for meta-learning papers [1, 2]. To ensure that the reader is familiar with at least some of the meta-environments, we included the in our benchmarks. The task distribution in the FwdBackw environments just has two tasks in its support, the tasks drawn during meta-training and meta-testing are identical. We fully agree, this is far away from optimal for evaluating the meta-generalization capabilities of the algorithms. Hence, we can only draw conclusions w.r.t. the meta-training performance in case of the FwdBackw environments. Finally, we emphasize the importance of better meta-RL benchmark environments and would highly welcome any work in this direction.
>
> [1] Chelsea Finn, Pieter Abbeel, Sergey Levine. Model-Agnostic Meta-Learning for Fast Adaptation of Deep Networks. ICML 2017.
> [2] Nikhil Mishra, Mostafa Rohaninejad, Xi Chen, and Pieter Abbeel. A Simple Neural Attentive Meta-Learner. In ICLR 2018.

---

### Public Comment · (anonymous) · 2018-11-19
**Gradient computation seems wrong in your codes for the variance of gradient experiments**

Given Hessian decomposition \nabla_\theta^2 J^{inner}(\theta) = H_1 + H_2 + H_{12} + H_{12}^T and your proposed approximation \E[\nabla_\theta^2 J^{LVC}(\theta)] = H_1 + H_2, what's the gradient should be evaluated? I believe that since you are arguing that you have a lower variance estimation of \nabla_\theta^2 J^{inner}(\theta) which is \E[\nabla_\theta^2 J^{LVC}(\theta)], the gradient to be evaluated should be the gradient resulting from Hessian-vector product, namely, the gradient used to update the original parameter of policy \theta.
What I see in your codes is you are actually evaluating variance of \nabla_\theta J^{LVC}(\theta), which is a mid-product of Hessian. It has no relation to the gradient of interest.

---

> ### Author Response · Authors · 2018-11-22
> **Computation of meta-gradient variance**
>
> Thanks a lot for your effort of reviewing and understanding our code! For experimentation purposes the code you refer to computes the gradients / gradient-variance of both the inner gradients and the meta-gradients. The respective statistics are logged in the lines 121-123 and 135-138 of meta_trainer_gradient_variance.py. What we report in Fig. 3 is the “Meta-GradientRStd” and corresponds to the meta-gradients which are used to update the original policy parameters \theta (as you suggested).  So, everything should comply with the experiment description in the paper.

---

### Author Response · Authors · 2018-11-29
**Updates in the paper**

Addressing the reviewers concerns and suggestions, we added further experimental results and explanations to the paper. In summary, the following changes have been made:

1) We extended the gradient variance experiments to more iterations and three environments. In accordance, we updated the respective experiment section.

2) We included DiCE into our performance benchmarks. Since there were already many curves in the benchmark figure, we split it into two figures. One figure with the full algorithms and the other with focus on the underlying gradient estimators
Fig. 2: ProMP, MAML-TRPO, E-MAML-TRPO, MAML-VPG
Fig. 3: LVC-VPG, DiCE-VPG, MAML-VPG, E-MAML-VPG

3) In section 5, we extended the explanation why the original RL-MAML implementation does not perform any pre-adaptation credit assignment

4) We fixed minor typos and notational inconsistencies throughout the paper

---

> ### Public Comment · (anonymous) · 2018-12-19
> **weak support to the claim**
>
> Hi authors,
>
> I used PyTorch to implement LVC on TRPO, and compare it with MAML+TRPO. It turns out that LVC has a lower variance and a worse average reward. When implement LVC on PPO, which has a name ProMP in your paper, it has a lower variance than MAML+TRPO and a higher average reward.
>
> This indicates that LVC can reduce variance, but the bias of LVC has a significant bad effect on performance. The reason ProMP has a better performance is probably because of the advantage of PPO over TRPO.
>
> I know you reported VPG based comparison, but VPG is well know for its instability, thus not a good testbed here.
> Even if so, it's hard to see LVC+VPG is better than MAML+VPG.
>
> I suggest you to report TRPO based results if you did experiments on it.

---

> > ### Author Response · Authors · 2018-12-21
> > **LVC-TPRO**
> >
> > Thanks a lot for implementing the proposed method in PyTorch and sharing your insights. For a subset of the environments we have results have results for LCV-TRPO. While LVC-TRPO seems to be better in in the FwdBack environments, it performs slightly worse than MAML-TRPO in the AntRandDir environment.
> > From a mathematical standpoint, MAML is more biased than LVC, because the H_1 RL-hessian term is missing in MAML when compared to the hessian estimates of LCV. We believe that one the following may explain the findings.
> >
> > Hypothesis 1: Since MAML doesn’t do any pre-adaptation credit assignment, i.e. the \nabla J_pre term is 0, the MAML meta-gradients are easier to estimate and might exhibit less variance. In meta-environments where task identification is trivial, pre-adaptation credit assignment plays a minor role and the higher variance of LVC might make it less sample-efficient compared to MAML. However, this hypothesis can hardly explain fact that LCV consistently outperforms MAML with VPG as outer optimizer.
> >
> > Hypothesis 2: As the monotonic policy improvement theory theory suggest, we must 1) account for changes in the pre-update action distribution and 2) bound changes in the pre-update state visitation distribution. Based on this, TRPO makes the maximally large step while fulfilling these conditions / constraints. However, as we point out in section 6, in Meta-RL, we have to fulfill the conditions for both the pre-adaptation and post-adaptation policy. If we just use TRPO as the outer optimizer, it will just fulfill 1) and 2) for the post-adaptation policy. As a result, the step and direction may be too large in order to suffice the conditions for the pre-adaptation policy. Since MAML ignores the pre-adaptation sampling distribution anyway, this would not be too problematic for MAML. But in case of LVC, this might hurt performance and outwage the benefits of LVC.
> >
> > In environments where task-identification is non-trivial (i.e. when we have non-dense rewards) such as in Fig. 5, pre-adaptation credit assignment plays a major role. In such environments, the average performance of MAML is far behind LVC. In general, the Meta-RL benchmarks we have right now in Fig. 3 are still pretty naive. It would be nice to see some more results in this direction with harder Meta-RL tasks etc. Please let us know if you have any experiment results in this direction and what you think about the hypotheses. As we have a proper way of interpreting the LCV-TRPO results, we are happy to include corresponding results in the paper. Thanks a lot for your support.

---

### Public Comment · ~Ankesh_Anand1 · 2018-12-10
**Typo / Indentation issue in Algorithm 1?**

 Steps 7-10 should occur for all the values of n right, not just n=0? Thus steps 7-10 might be need to unindented by one level.

---

> ### Author Response · Authors · 2018-12-10
> **Algorithm 1**
>
> Thank you for your comment.
>
> One of the main advantages of ProMP is that it can perform multiple meta-gradient steps without re-sampling trajectories. The mechanisms in the ProMP objective (likelihood ratio + clipping + KL penalty) stabilize the meta-optimization and ensure that no policy collapse happens when doing multiple gradient steps with the same data.  This makes the algorithm more sample-efficient and faster in compute time.
>
> Hence, the indentation of step 7-10 in algorithm 1 is intended since we only want to sample trajectories once and then perform N meta-gradient steps with it. In practice, we usually set N=3 or N=5. Due to page constraints in the paper, we may have explained this only insufficiently. We aim to better clarify this in the camera-ready version of the paper.

---

### Public Comment · (anonymous) · 2022-02-11
**Questions about Figure 4-5**

A typo: Equation(2) should log\pi\theta instead of log\pi\theta'
Q1: Figure 4, why are the verticle axis of the top three plots the average return but the title is gradient variance?
Q2: In Figure 5,
    In every method, is the pre-update policy just the random policy or the policy after meta-training?
    Are all trajectories of post-update taken after three inner adaptation steps?
Q3: I am wondering about the advantage of multiple outer updates. As described in Algorithm 1,  a full update contains one inner update + multiple outer updates (meta-update) without re-sampling. However, I think multiple outer updates with the same sample  = one outer update with a larger learning rate, right?

---

### Meta-Review · Area_Chair1 · 2018-12-14
**Interesting work, novel contribution**

**Confidence:** 4
**Recommendation:** Accept (Poster)

**Metareview:**

The paper studies the credit assignment problem in meta-RL, proposes a new algorithm that computes the right gradient, and demonstrates its superior empirical performance over others.  The paper is well written, and all reviewers agree the work is a solid contribution to an important problem.